# Classifiers are Forgetful! Balancing the Mutual Causal Effects in Class-Incremental Learning

## Abstract

Class-Incremental Learning (CIL) is a practical and challenging problem for achieving general artificial intelligence. Pre-Trained Models (PTMs) have recently led to breakthroughs in both visual and natural language processing (NLP) tasks. Despite recent studies showing PTMs' potential ability to learn sequentially, a plethora of work indicates the necessity of alleviating the catastrophic forgetting of PTMs. Through a pilot study and a causal analysis of CIL, we reveal that the problem lies in the imbalanced effect between new and old data, which leads to the forgetting of classifiers. To alleviate this problem, we propose BaCE, a method retrieving the causal effects from new data to the adaptation of old classes and from old data to the adaptation of new classes. By balancing the causal effects, BaCE enables the causal effects from new and old data to help the adaptation to each class. We conduct extensive experiments on three different tasks (Image Classification, Text Classification, and Named Entity Recognition) with various backbones (ResNet-18, ViT, BERT) in the CIL setting. Empirical results show the proposed method outperforms a series of CIL methods on different tasks and settings. The code will be publicly available after acceptance [1].

## 1 Introduction

Incremental Learning (IL) aims at endowing machine learning systems with the ability to continuously learn novel concepts, which is critical for the research on human-level intelligence. This paper focuses on Class-Incremental Learning (CIL), the most challenging and practical scenario in IL (Prabhu et al., 2020; Buzzega et al., 2020; Van de Ven & Tolias, 2019). CIL requires models to classify all classes seen so far without task indexes. Therefore, catastrophic forgetting (French, 1999; Rosenstein et al., 2005; McCloskey & Cohen, 1989) may occur within tasks and between tasks (Tao et al., 2023). While a good number of approaches (Kirkpatrick et al., 2017; Hu et al., 2021; Rebuffi et al., 2017; Hou et al., 2019; Wu et al., 2019) have been proposed in recent years, most of them rely heavily on experience replay (Chaudhry et al., 2019), and they suffer from substantial performance deterioration when the replay data is limited or even non-existent.

Recently, Pre-Trained Models (PTMs), especially pre-trained Transformers (Vaswani et al., 2017), have achieved remarkable progress in both computer vision (He et al., 2022; Dosovitskiy et al., 2020) and natural language processing (NLP) (Devlin et al., 2019; OpenAI, 2023). Despite its success across various benchmarks, the CIL ability of pre-trained Transformers is yet to be fully explored and understood. On the one hand, the CIL performance of PTMs (Wang et al., 2022e; Huang et al., 2021; Zheng et al., 2022; de Masson D'Autume et al., 2019) is still far from satisfactory with limited buffer data. On the other hand, Ramasesh et al. (2022); Tao et al. (2023) show that PTMs are inherently resilient to catastrophic forgetting even without buffer data. This contradictory phenomenon urges us to explore the reason behind it.

First, we conduct a pilot study based on linear probing (Tao et al., 2023; Chen et al., 2023) in CIL settings. In our linear probing study, the backbone of PTMs (i.e., the encoder) is frozen while the classifier is re-trained on the data from all classes learned so far. Surprisingly, we find that simply

---

[1]Anonymous URL: https://anonymous.4open.science/r/BaCE-F055

re-training the classifier improves the average accuracy (Wang et al., 2022e; Chaudhry et al., 2019) from 14.1% to 83.2% without buffer data and from 60.9% to 84.5% with 100 buffer samples in the 20-step setting of split CIFAR-100 (Krizhevsky et al., 2009). In other words, pre-trained encoders are capable of learning new classes without much forgetting, but the classifier forgets how to classify them.

In light of the fact that classifiers are usually randomly initialized while encoders are endowed with prior knowledge during pretraining, classifiers may learn new knowledge in a different manner. Then, we track the distance of class centers between encoders and classifiers. We find that when models adapt to each new task, new class centers of classifiers always align with those of encoders. In stark contrast, old class centers of classifiers are always pushed away from those of encoders. This finding indicates that the effects of adapting to new and old classes are contradictory. Because pre-trained encoders are more resilient to forgetting, the confrontation phenomenon between new and old classes leads to the forgetting of classifiers. Moreover, we discover that the phenomenon also exists when the encoder is ResNet-18 (He et al., 2016), indicating it may be prevalent in CIL.

To further analyze this problem, we introduce the causal graph: a graphical framework that stands in the cause-effect interpretation of the data, but not merely the statistical association of them (Glymour et al., 2016; Pearl, 2009). Specifically, by framing the data, features, and models into causal graphs, we find that the community has overlooked two causalities in CIL: (1) the causal effect of new data on learning old classes; (2) the causal effect of old data on learning new classes. Without these two effects, only new/old data has a causal effect on adapting to new/old classes, which hinders the adaptation of old/new classes. In other words, the effects of new and old data enhance the adaptation to new and old classes separately but impede the learning of the other classes. To this end, we propose *Balancing Causal Effects* (**BaCE**), a method that encourages new and old data mutually helps model adaptation for mitigating the catastrophic forgetting in classifiers.

Finally, we conduct extensive experiments on three CIL tasks: Continual Image Classification, Continual Text Classification, and Continual Named Entity Recognition. The experimental results suggest that BaCE outperforms a series of CIL methods based on ResNet (He et al., 2016), e.g.Rebuffi et al. (2017); Hou et al. (2019); Wu et al. (2019); Hu et al. (2021), Vision Transformers (Dosovitskiy et al., 2020), e.g.,Wang et al. (2022e), and BERT (Devlin et al., 2019), e.g.,de Masson D'Autume et al. (2019); Huang et al. (2021); Zheng et al. (2022); Wang et al. (2022b).

In summary, our contributions are three-fold: (1) We find that the confrontation phenomenon leads to the forgetting of classifiers, resulting in models suffering from catastrophic forgetting seriously even when encoders preserve old knowledge. (2) We delve into the causalities in CIL and reveal that the reason for the confrontation phenomenon lies in the imbalanced causal effects between new and old data. To address this, we propose BaCE to balance the causal effects when learning each category, which enables models to learn new and old data jointly. (3) We conduct experiments on both visual and NLP tasks to verify the effectiveness of BaCE. The result indicates that BaCE mitigates the confrontation phenomenon and outperforms alternative CIL methods by a large margin.

## 2 RELATED WORK

We summarize six parts of related work: Class-Incremental Learning (A.1), Incremental Learning with PTMs (A.2), Probing Study in Incremental Learning (A.3), Imbalanced Problem in CIL (A.4), Causal Inference in CV and NLP (A.5), and Continual Causal Discovery (A.6). The full related work is in the Appendix A.

**Class-Incremental Learning.** Existing CIL methods can be roughly divided into three groups: regularization-based methods, exemplar-based methods, and architecture-based methods. *Regularization-based* methods estimate the importance of parameters for previous tasks and penalize the update of important parameters for mitigating forgetting (Kirkpatrick et al., 2017; Li & Hoiem, 2017; Zenke et al., 2017). These methods did not achieve satisfactory performance under challenging and complex scenarios (Rebuffi et al., 2017). *Exemplar-based* methods store representative instances from old classes and replay the stored instances when learning new tasks (Rebuffi et al., 2017; Hou et al., 2019; Wu et al., 2019; Buzzega et al., 2020; Arani et al., 2022). Although they achieve state-of-the-art performance on various CIL benchmarks (Chaudhry et al., 2019), their performance typically deteriorates when the buffer size is small. More importantly, over-reliance on exemplars

violates the setting of CIL and simplifies CIL as Multi-Task Learning (MTL). This study investigates practical CIL scenarios where buffer size is limited or without rehearsal buffer. *Architecture-based* methods increase model components incrementally to meet the requirements of new classes (Serra et al., 2018; Rajasegaran et al., 2019; Yan et al., 2021; Kim & Han, 2023). However, these models require large memory when there are many tasks. Furthermore, architecture-based methods implicitly introduce an extra memory budget since the backbones from history are treated as unforgettable checkpoints (Zhou et al., 2023a).

**Incremental Learning with PTMs.** Most existing Incremental Learning (IL) methods are based on Convolutional Neural Networks (CNN). Recently, IL with PTMs has become a newly emerged research direction. For example, Wang et al. (2022e;d) leverage prompt (Liu et al., 2023) for IL and achieve superior performance even without replay data. However, Wang et al. (2022e;d) introduce an extra architecture called prompt pool, which may implicitly serve as an extra memory for preserving old knowledge. Different from Wang et al. (2022e;d), BaCE does not rely on extra components and is applicable to various backbones. Besides, Ermis et al. (2022); Razdaibiedina et al. (2023) utilize adapter (Houlsby et al., 2019) and prompt respectively to solve Task-Incremental Learning (TIL), which is an easier scenario than CIL since the task indexes are given during inference. Wang et al. (2022c) learns prompts independently across domains for Domain-Incremental Learning. Moreover, Ke et al. (2022b;a); Jang et al. (2022) focus on continual pretraining with PTMs, which is a more general scenario of continual learning.

## 3 A Pilot Study for CIL with PTMs

Typically, a model can be divided into two components: a feature encoder and a task-specific classifier. Without loss of generality, we use PTMs (without original classification heads) as the feature encoder and a linear layer with cosine normalization (Hou et al., 2019) as the classifier. This section will clarify the key to PTMs' forgetting. The training setting is in the Appendix B.1.

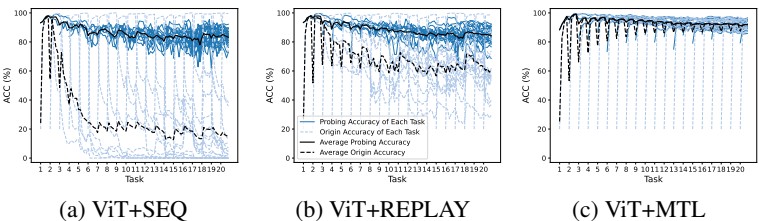

| (a) ViT+SEQ | (b) ViT+REPLAY | (c) ViT+MTL |

Figure 1: The probing study on the 20-step split CIFAR-100. We use ViT-B/16 (ViT) pretrained on ImageNet-21k (Dosovitskiy et al., 2020) as backbones. The buffer size is 200 in REPLAY. The blue curve represents the accuracy on each task, and the black curve represents the average accuracy over the tasks learned so far. The solid and the dotted line represent the probing and original average accuracy, respectively.

### 3.1 Probing Study

The linear probing (Tao et al., 2023; Chen et al., 2023) is a commonly used technique to measure the encoding ability of feature encoders. In our probing study, we aim to probe each model checkpoint in CIL. Specifically, we fix the encoder of each checkpoint and re-train its classifier on the data of all tasks that have been learned so far. In this way, we obtain the probing performance of each checkpoint, and this performance can be regarded as the upper limit performance when classifiers do not forget (Fig. 3a). Correspondingly, we call the performance of models with original classifiers as *the original performance* (B.2).

We consider three methods for probing: sequential training (SEQ), experience replay (REPLAY), and multi-task learning (MTL). The result of the probing study on split CIFAR100 (Krizhevsky et al., 2009) is shown in Fig. 1. Fig. 1 shows that pre-trained encoders are resistant to forgetting while trained-from-scratch classifiers are prone to forgetting. Besides, more rehearsal data helps close the gap between original and probing performance (B.3). Furthermore, we also conduct a

probing study on ResNet-18 (He et al., 2016), and the result in the Appendix B.4 shows similar trends. Although the probing performance of ResNet-18 is lower than ViT, the gap between probing and original performance is prominent in both SEQ and REPLAY settings. Therefore, the probing study on ResNet-18 and ViT reveals a prevalent phenomenon in CIL: classifiers forget at a much faster speed than encoders.

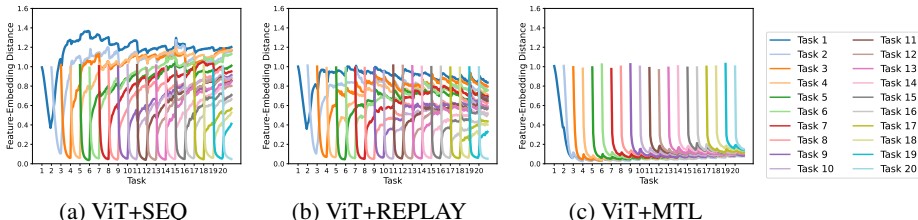

| (a) ViT+SEQ | (b) ViT+REPLAY | (c) ViT+MTL |
|---|---|---|

Figure 2: The evolution of *feature-embedding distance*. The backbone model is ViT-B/16, and the dataset is the 20-step split CIFAR-100. Each colour represents the average *feature-embedding distance* of classes from an incremental task.

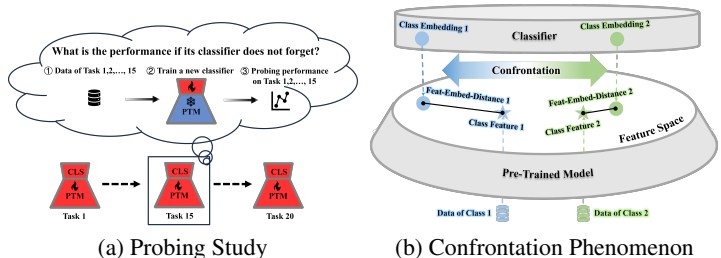

| (a) Probing Study | (b) Confrontation Phenomenon |
|---|---|

Figure 3: The illustration of (a) the probing study and (b) the confrontation phenomenon in the tracking study.

## 3.2 TRACKING STUDY

The probing study shows that classifiers and encoders always forget at a different speed. To understand why it happens, we track the learning process of classifiers and encoders from a feature-level perspective. In classifiers, the representation of each class (*class embedding*) corresponds to a row vector in the weight matrix. In encoders, the representation of each class (*class feature*) can be estimated as the average feature of all training samples from that class. Intuitively, the feature-embedding distance of one class is small when models learn how to distinguish it from others, and the feature-embedding distance of one class is large when models fail to do so.

We use the same settings in the probing study and track the feature-embedding distance in the whole learning process. Fig. 2 shows that the feature-embedding distance of each task (each curve) decreases to a small value when this task is newly adapted, and the distance increases as models learn more tasks. For each CIL step, new classes are learned as their feature-embedding distances are minimized. In contrast, old classes are forgotten as their feature-embedding distances grow. In other words, models always align new class embeddings with the corresponding new class features while simultaneously pushing old class embeddings away from the corresponding old class features. Similarly, new classes will be forgotten if new models are trained only on old data. Therefore, the new/old data hinders the adaptation of old/new classes, and we call it the *confrontation phenomenon* in this paper (Fig. 3b). Alleviating the confrontation phenomenon is important because it may hinder models from learning the optimal representations of both new and old data. It is worthy noting that the confrontation phenomenon is different from the class-imbalanced problem since the former describes the whole learning process at the feature level, while the latter describes the prediction bias at inference time. Please refer to the Appendix A.4 for more details.

Naturally, we can adopt the average feature-embedding distance of all tasks as an indicator of the degree of the confrontation phenomenon. With the availability of more replay data, we find that (i) the confrontation phenomenon is alleviated (Fig. 2) and (ii) the performance gap between probing and original performance becomes smaller (Fig. 1). It indicates that we may close the performance gap by alleviating the confrontation phenomenon (more details are in the Appendix B.3). We summarize the findings in the probing and tracking study in the Appendix B.5.

How can we alleviate the confrontation phenomenon? Obviously, storing more old data is a straightforward solution. But how can we achieve this when storing limited old samples or even no old samples? Recall that the confrontation phenomenon is caused by the effect of the adaptation process of other tasks. Can we alleviate the confrontation phenomenon by encouraging models to adapt to new and old tasks with "good" effects from both new and old data? To answer this question, we need to first sort out the causal relationships in CIL.

## 4 METHODOLOGY

### 4.1 REVISITING THE CAUSALITIES IN CIL

Formally, the goal of CIL is to learn a single model $f_\theta : \mathbf{x} \to y \in \mathcal{Y}$ from the sequence of tasks $\mathcal{D}$ $\mathcal{D} = \{\mathcal{D}_1, \mathcal{D}_2, \cdots, \mathcal{D}_T\}$, where the $t$-th task $\mathcal{D}_t = \{(\mathbf{x}_i^t, y_i^t)\}_{i=1}$ contains input samples $\mathbf{x}_i^t \in \mathcal{X}_t$ and labels $y_i^t \in \mathcal{Y}_t$. The label sets of different tasks are exclusive: $\mathcal{Y}_1 \cap \mathcal{Y}_2 \cdots \mathcal{Y}_T = \emptyset$. When adapting to each new task, the classifier expands the output dimension for predicting new categories. In the data replay setting, a buffer $\mathcal{M}$ is introduced for storing old representative instances.

Each CIL step can be framed into a causal graph (Pearl, 2009), where nodes are variables and directed edges represent the causalities between variables. Fig. 4a is the causal graph of SEQ. In Fig. 4a, $X^{old}, X^{new}$ are the input samples from old and new tasks, and $H^{old}, H^{new}$ are the extracted features, respectively. $Z$ represents output logits, i.e., the model predictions before *softmax* layer. The superscript *old* and *new* of $Z$ represents they are computed from $X^{old}$ and $X^{new}$ respectively. Moreover, the subscript *[old]* and *[new]* represent the logits over the category from old and new tasks. Since both features and class embeddings determine logits, optimizing the logits of new ($Z_{[new]}$) and old classes ($Z_{[old]}$) encourages the adaptation of new and old classes, respectively. Although $X^{new}$ have effects on both $Z_{[new]}^{new}$ and $Z_{[old]}^{new}$ in forward propagation, only the causal path $X^{new} \to H^{new} \to Z_{[new]}^{new}$ helps models adapt to new classes while the other path $X^{new} \to H^{new} \to Z_{[old]}^{new}$ hinders this process. The causal graph of REPLAY is shown in Fig. 4b, where $X^{buf}$ is the rehearsal samples selected from $X^{old}$. Similarly, only the causal path $X^{old} \to X^{buf} \to H^{buf} \to Z_{[old]}^{buf}$ adapts models to old classes.

### 4.2 BALANCING THE CAUSALITIES IN CIL

In both SEQ and REPLAY settings, the effect of adapting to new (old) classes is imbalanced since only $X^{new}$ ($X^{old}$) contributes to the adaptation of new (old) classes. To address this problem, we propose BaCE, which balances the effects from $X^{new}$ and $X^{old}$ when adapting to each class. We illustrate the difference between SEQ, REPLAY, and BaCE in Fig.4f.

*Effect*$_{old}$: **Learning Old Classes with Balanced Causal Effects from $X^{old}$ and $X^{new}$.** Knowledge distillation (Hinton et al., 2015) has been proven to be effective in CIL. Following Wu et al. (2019); Hou et al. (2019); Buzzega et al. (2020); Li & Hoiem (2017), we use the model trained on previous tasks (denoted as $f^{t-1}$) as the teacher and the model of the current CIL step (denoted as $f^t$) as the student for knowledge distillation.

When not using data replay, we define *Effect*$_{old}$ as follows:

$$\max Effect_{old} = \mathbb{E}_{(x,y)\sim\mathcal{D}_t}(-\alpha\mathcal{L}_{KL}(S_{[old]}^{new}(x), S0_{[old]}^{new}(x)) \tag{1}$$

$\mathcal{L}_{KL}(\cdot, \cdot)$ is the Kullback-Leibler Divergence. $(x, y)$ is sampled from $\mathcal{D}_t$. $S_{[old]}^{new}$ and $S0_{[old]}^{new}$ are the score of old classes output by $f^t$ and $f^{t-1}$. $\alpha$ is the scaling hyper-parameters. The causal graph of *Effect*$_{old}$ is shown in Fig. 4c and its rationale is as follows: $H_0^{new}$ is the feature of $X^{new}$ extracted by $f^{t-1}$. $H_0^{new}$ is also determined by $X^{old}$ due to the fact that $f^{t-1}$ is trained on $X^{old}$.

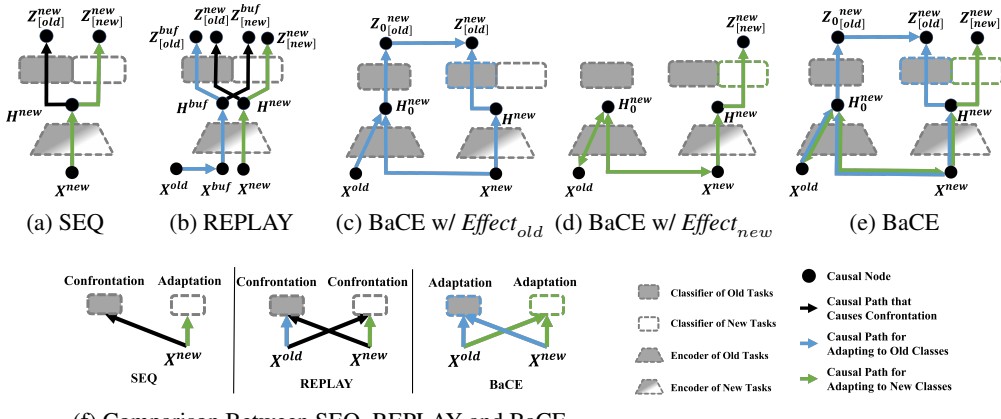

(a) SEQ  (b) REPLAY  (c) BaCE w/ $Effect_{old}$  (d) BaCE w/ $Effect_{new}$  (e) BaCE

(f) Comparison Between SEQ, REPLAY and BaCE

Figure 4: The Causal Graph of SEQ, REPLAY, and BaCE (Ours) with $Effect_{old}$ and $Effect_{new}$ in each CIL step. The directed edges represent the causal effects between variables in the *forward propagation*. The blue and green paths represent the causal effects of adapting to *old* and *new* classes, respectively, when taking *back propagation* into consideration.

$Z_0{}^{new}_{[old]} \rightarrow Z^{new}_{[old]}$ is built by minimizing Eq. 1. In summary, we build the following causal paths from $X^{old}$ and $X^{new}$ to $Z^{new}_{[old]}$ for adapting to old classes: $X^{old} \rightarrow H_0^{new} \rightarrow Z_0{}^{new}_{[old]} \rightarrow Z^{new}_{[old]}$, $X^{new} \rightarrow H_0^{new} \rightarrow Z_0{}^{new}_{[old]} \rightarrow Z^{new}_{[old]}$ and $X^{new} \rightarrow H^{new} \rightarrow Z^{new}_{[old]}$. We provide detailed proof of this conclusion in the Appendix C.1.

When using data replay, we follow DER++ Buzzega et al. (2020) to enhance the learning of old classes by optimizing the classification and the distillation targets on old data. $Effect_{old}$ can be enhanced as follows:

$$Effect_{old-replay} = Effect_{old} + \mathbb{E}_{(x,y)\sim\mathcal{M}}(-\mathcal{L}_{CE}(x,y) - ||Z^{buf}_{[old]}(x) - Z_0{}^{buf}_{[old]}(x)||_2^2) \qquad (2)$$

$\mathcal{L}_{CE}(\cdot,\cdot)$ is the cross-entropy loss. $Z^{new}_{[old]}$ and $Z_0{}^{new}_{[old]}$ are the logits of old classes output by $f^t$ and $f^{t-1}$. $||\cdot||_2$ is the Euclidean distance. We note that the proposed causal graphs in Fig. 4c are independent of data replay since we consider $X^{old}$ and $X^{buf}$ separately. It is worth noting that BaCE is independent of DER++, which is considered a baseline in our experiments. We highlight that the knowledge distillation term in Eq. 1 is crucial from the causal perspective and should not be discarded as in DER++. We empirically show $Effect_{old}$ brings considerable improvement based on DER++.

***Effect*new: Learning New Classes with Balanced Causal Effects from $X^{new}$ and $X^{old}$.** Since models are always strongly biased toward new classes, mitigating the confrontation phenomenon in learning new classes is necessary. We propose to balance the effect on learning new classes by building causal paths from both $X^{new}$ and $X^{old}$ to $Z_{[new]}$ (denoted as $Effect_{new}$).

If we ignore the directionalities in Fig. 4c, there is a path from $X^{old}$ to $Z^{new}_{[new]}$: $X^{old} \rightarrow H_0^{new} \leftarrow X^{new} \rightarrow H^{new} \rightarrow Z^{new}_{[new]}$. If $X^{new}$ is influenced by $X^{old}$, $X^{old}$ will have an effect on $Z^{new}_{[new]}$ and thus helps adapt to new classes. Thanks to causal inference, we can achieve this by conditioning the *collider* $H_0^{new}$. Specifically, $H_0^{new}$ is the joint outcome of the independent causes $X^{old}$ and $X^{new}$. Interestingly, once the common effect $H_0^{new}$ is observed, the causes $X^{old}$ and $X^{new}$ become dependent on each other [2]. We would like to provide an example (Pearl, 2009) for clarifying this phenomenon: Suppose a school's admission criteria require high grades or special athletic talents. In that case, these two attributes will be found to be negatively correlated in the school's student population, even if these attributes are not related throughout the entire population. By conditioning on $H_0^{new}$, a bidirectional causal path $X^{old} \leftrightarrow H_0^{new} \leftrightarrow X^{new}$ is built as shown in Fig. 4d. We call this effect the collider effect since it holds only when the collider $H_0^{new}$ is observed.

---

[2]This phenomenon is also known as *Berkson's paradox* in (Berkson, 1946) and as the *explaining away effect* in (Peari & Kim, 1983).

By utilizing the collider effect, $Effect_{new}$ is estimated as follows:

$$Effect_{new} = -\mathbb{E}_{(x,y)\sim\mathcal{D}_t}\mathcal{L}_{CE}(\sum_{\tilde{x}\in\{x\}\cup\mathcal{N}(x)} W(\tilde{x},x)S(\tilde{x}),y), \tag{3}$$

where $\sum_{\tilde{x}\in x\cup\mathcal{N}(x)} W(\tilde{x},x)=1$; $\mathcal{N}$ is the set of K-Nearest-Neighbors (KNNs) in the feature space of $f^{t-1}$; $S(\tilde{x})$ is the score prediction of $\tilde{x}$; $W(\tilde{x},x)$ is the weight of $S(\tilde{x})$ and it is defined as follows:

$$W(\tilde{x},x) = \begin{cases} W_0, & \text{when} \quad \tilde{x}=x; \\ \frac{(1-W_0)/||H_0(\tilde{x})-H_0(x)||_2}{\sum_{\tilde{x}'\in\mathcal{N}(x)} 1/||H_0(\tilde{x}')-H_0(x)||_2}, & \text{otherwise.} \end{cases} \tag{4}$$

In Eq. 4, we define the weight of each neighbour as the normalized reciprocal of Euclidean distance to the input sample. Eq. 3 is the same as the standard classification loss on new data, except that scores are computed as the weighted sum of the score of input samples and their neighbours. In other words, we estimate the score of a sample $x$ as the joint score of the sample $x$ itself as well as its KNNs $\mathcal{N}(x)$. In the Appendix C.3, we provide detailed derivation and further explanation of Eq. 3. From the causal perspective, $X^{old}$ has causal effect on $Z^{new}_{[new]}$ through the path $X^{old}\leftrightarrow H_0^{new}\leftrightarrow X^{new}\rightarrow H^{new}\rightarrow Z^{new}_{[new]}$. Therefore, maximizing $Effect_{new}$ encourages models to adapt to new classes with causal effects from both $X^{new}$ and $X^{old}$.

**Overall Objective of BaCE.** To sum up, the overall objective is given as follows: When rehearsal buffer is unavailable, BaCE maximizes $Effect = Effect_{new} + Effect_{old}$; When rehearsal buffer is available, BaCE maximizes $Effect_{replay} = Effect_{new} + Effect_{old-replay}$. Besides, we propose to update the teacher model $f^{t-1} = \beta f^{t-1} + (1-\beta)f^t$ every training epoch to facilitate the adaptation to new data distribution. We empirically find that $\beta = 0.9$ yields better performance. Besides, we compare BaCE with prior works and summarize the algorithm in the Appendix C.2, C.4, C.5.

**Why BaCE mitigates the confrontation phenomenon?** In the REPLAY setting, the learning objectives of new and old data are inherently contradictory. When the buffer size is limited, the contradictory objectives hinder the adaptation to both new and old classes, and it results in the confrontation phenomenon. BaCE fundamentally overcomes this problem because BaCE encourages model adaptation on new and old data in a collaborative manner (Fig. 4e). Further discussion is provided in the Appendix C.6 and C.7.

## 5 EXPERIMENTS

To verify the effectiveness of BaCE, we conduct experiments on three tasks: Continual Image Classification (Continual IC), Continual Text Classification (Continual TC), and Continual Named Entity Recognition (Continual NER). Due to the space limitation, additional empirical results (e.g., hyper-parameter analysis, runtime analysis, experiments of ResNet-18, and the evolution of average accuracy) on Continual IC and all experiments on Continual TC and Continual NER are provided in the Appendix D.2, and D.3.

### 5.1 EXPERIMENTAL SETTINGS

**Training and Evaluation.** We use CIFAR-100, CIFAR-10 (Krizhevsky et al., 2009), 5-datasets (Ebrahimi et al., 2020), OminiBenchmark, Tiny-ImageNet, ObjectNet, ImageNet-R, VTAB in this paper. The introduction and statistics are provided in Appendix D.1 and Table 5. We use ViT-B/16, ViT-B/16-IN21K (Dosovitskiy et al., 2020), DeiT-S/16 Touvron et al. (2021) and ResNet-18 He et al. (2016) as the backbone. We adopt three widely-used metrics for evaluation: *Average Accuracy* (AverACC) (Chaudhry et al., 2019), *Forgetting* (FGT) (Chaudhry et al., 2018) and *Forward Transfer* (FWT) (Chaudhry et al., 2018). The AverACC refers to the average accuracy after learning the final task. In $Effect_{new}$, we set the number of neighbors $K = 5$ and the weight $W_0 = 0.95$. In $Effect_{old}$, we set $\alpha = 5$ when no buffer is available and $\alpha = 1$ when the buffer is available. The detailed training settings and hyper-parameter analysis are in the Appendix D.1.

**Baselines** We consider the following competitive CIL methods: Experience Replay (ER), LwF (Li & Hoiem, 2017), EWC (Kirkpatrick et al., 2017), BiC (Wu et al., 2019), LUCIR (Hou et al., 2019), PODNET (Douillard et al., 2020), DDE (Hu et al., 2021), DER++ (Buzzega et al., 2020), L2P (Wang

et al., 2022e), Co2L (Cha et al., 2021), Gdumb (Prabhu et al., 2020), CLSER (Arani et al., 2022), FOSTER (Wang et al., 2022a), MEMO(Zhou et al., 2023a), BEEF (Wang et al., 2023). Sequential Training (SEQ) and Multi-Task Learning (MTL) are CIL's lower and upper bounds. We load the same backbone model for all baselines and our method. A detailed introduction and experimental settings are in the Appendix D.1.

## 5.2 RESULTS AND ANALYSIS

**Comparison with Baselines.** The comparison between BaCE and various CIL baselines is shown in Table 1 and 2. All methods use ViT-B/16 as the backbone. "OOM" refers to Out-Of-GPU memory. We provide the full result with standard derivations and FWT on 5-/10-/20-step CIFAR100 and 5-datasets in the Appendix D.1. When replay data is unavailable, BaCE outperforms the regularization-based method EWC and the distillation-based method LwF significantly. The result indicates that simply constraining the update of model parameters fails to utilize the inherent ability of PTMs. Besides, it may be susceptible to negative transfer (Chen et al., 2019). When replay data is available, BaCE performs better than a series of competitive CIL methods. The result also shows that BaCE has a lower FGT and a higher FWT compared with other methods, indicating that balancing the causal effects is beneficial to preserving old knowledge and learning new concepts.

Table 1: The comparison with baselines on CIFAR-100.

| Buffer Size | Method | CIFAR100 (10 step) | |
|---|---|---|---|
| | | AverACC (↑) | FGT (↓) |
| 0 | SEQ | 24.09 | 80.85 |
| | LwF | 45.88 | 51.93 |
| | EWC | 29.28 | 75.26 |
| | BaCE w/o $Effect_{new}$&$Effect_{old}$ | 23.84 | 81.58 |
| | BaCE w/o $Effect_{new}$ | 46.03 | 50.72 |
| | BaCE w/o $Effect_{old}$ | 29.43 | 72.22 |
| | BaCE (Ours) | **51.84** | **32.99** |
| 500 | ER | 70.78 | 30.28 |
| | BiC | 74.59 | 24.84 |
| | LUCIR | 74.52 | 21.68 |
| | PODNET | 48.29 | 55.42 |
| | DDE | 72.02 | 28.43 |
| | DER++ | 75.17 | 25.96 |
| | CLSER | 78.54 | 19.68 |
| | FOSTER | 76.84 | / |
| | MEMO | 78.96 | 17.65 |
| | BEEF | OOM | OOM |
| | BaCE w/o $Effect_{new}$&$Effect_{old}$ | 75.45 | 24.91 |
| | BaCE w/o $Effect_{new}$ | 82.13 | 17.82 |
| | BaCE w/o $Effect_{old}$ | 78.60 | 20.87 |
| | BaCE (Ours) | **84.59** | **12.58** |
| ∞ | MTL | 91.25 | / |

Table 2: The comparison with baselines on 5-datasets. †: Results from Wang et al. (2022e).

| Buffer Size | Method | 5-datasets (5 step) | |
|---|---|---|---|
| | | AverACC (↑) | FGT (↓) |
| 0 | FT-seq-frozen † | 39.49 | 42.62 |
| | FT-seq † | 20.12 | 94.63 |
| | EWC † | 50.93 | **34.94** |
| | LwF † | 47.91 | 38.01 |
| | BaCE w/o $Effect_{new}$&$Effect_{old}$ | 21.37 | 95.45 |
| | BaCE w/o $Effect_{new}$ | 50.61 | 47.49 |
| | BaCE w/o $Effect_{old}$ | 30.69 | 75.64 |
| | BaCE (Ours) | **54.99** | 37.79 |
| 500 | ER † | 84.26 | 12.85 |
| | Gdumb † | 70.76 | / |
| | BiC † | 85.53 | 10.27 |
| | DER++ † | 84.88 | 10.46 |
| | Co2L † | 86.05 | 12.28 |
| | L2P † | 88.95 | **4.92** |
| | CLSER | 89.43 | 6.20 |
| | FOSTER | 74.96 | / |
| | MEMO | 89.59 | 5.37 |
| | BEEF | 79.13 | / |
| | BaCE w/o $Effect_{new}$&$Effect_{old}$ | 86.20 | 9.16 |
| | BaCE w/o $Effect_{new}$ | 88.58 | 5.64 |
| | BaCE w/o $Effect_{old}$ | 88.79 | 6.20 |
| | BaCE (Ours) | **89.80** | 5.22 |
| ∞ | MTL † | 93.93 | / |

**Ablation Study.** We consider three ablated versions of BaCE: (1) BaCE w/o $Effect_{new}$: we substitute the objective in $Effect_{new}$ as traditional cross-entropy loss on new data. (2) BaCE w/o $Effect_{old}$: we remove the objective $Effect_{old}$. (3) BaCE w/o $Effect_{new}$&$Effect_{old}$: combining (1) and (2). When buffer size is zero, $Effect_{old}$ is crucial for preserving old knowledge. We note that even when using data replay, $Effect_{old}$ inflates the performance by exploiting the causal effects of new data, which is overlooked by prior works (Buzzega et al., 2020). Besides, $Effect_{new}$ brings considerable improvements under various buffer size settings, suggesting that introducing the old data effect to learning new classes alleviates the confrontation phenomenon and reduces the forgetting of old knowledge.

**Combined with Other Potential Solutions to the Confrontation Phenomenon.** There are other potential solutions for addressing the confrontation phenomenon: (1) Fixing encoders (Fix_Enc); (2) Fixing old classifiers (Fix_Cls); (3) Fixing both encoders and old classifiers (Fix_Enc_Cls); (4) Initializing the new classifiers with imprinted weights (Qi et al., 2018) (Imprinted Weights). We train each method on the 20-step split CIFAR-100 with ViT-B/16, and the buffer size is 500. Table 3 shows that Fix_Enc_Cls and Imprinted Weights improve ER. When we combine these two methods with BaCE, BaCE+Imprinted Weights achieves superior performance. In contrast, Fix_Enc_Cls degrades the performance of BaCE. The reason may be that Fix_Enc_Cls limits the potential forward/backward transfer between tasks.

Table 3: Potential solutions to the confrontation phenomenon.

| Method | AverACC (↑) | Δ AverACC | Feat-Embd-Dist (↓) | Δ Feat-Embd-Dist |
|---|---|---|---|---|
| ER | 70.78 | / | 0.24 | / |
| ER+Fix_Enc | 57.10 | -13.68 | 0.68 | +0.44 |
| ER+Fix_Cls | 24.59 | -46.19 | 0.46 | +0.22 |
| ER+Fix_Enc_Cls | 72.73 | +1.95 | 0.20 | -0.04 |
| ER+Imprinted Weights | 71.62 | +0.84 | 0.18 | -0.06 |
| BaCE w/o $Effect_{new}$&$Effect_{old}$ | 75.45 | +4.67 | 0.18 | -0.06 |
| BaCE w/o $Effect_{new}$ | 82.13 | +11.35 | 0.14 | -0.10 |
| BaCE w/o $Effect_{old}$ | 78.60 | +7.82 | 0.18 | -0.06 |
| BaCE (Ours) | 84.59 | +13.81 | 0.13 | -0.11 |
| BaCE+Fix_Enc_Cls | 74.40 | +3.62 | 0.19 | -0.05 |
| BaCE+Imprinted Weights | **84.77** | **+13.99** | **0.13** | **-0.11** |

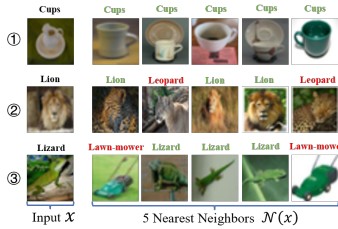

Figure 5: The KNNs in $Effect_{new}$

**Visualization of K-Nearest-Neighbors in $Effect_{new}$.** Recall that Eq. 3 estimates the collider effect as the cross-entropy loss of the joint score of an input sample $x$ and its neighbors $\mathcal{N}(x)$ in the feature space of $f^{t-1}$. Fig. 5 provides three examples to demonstrate how $X^{old}$ affects $X^{new}$ through $H_0^{new}$. (More examples are in the Appendix D.1) The ground-truth label is on the top of each image. The green and red labels of neighbors represent whether or not they are the same as the input sample. In the first example, the ground-truth category of the input sample (i.e., cups) is the same as those of neighbors. It indicates that the teacher model $f^{t-1}$ can recognize new classes with prior knowledge before training on them. Although some neighbors may have different categories from those of input samples (e.g., the latter two examples), input samples and their neighbors bear a resemblance in the feature space of the teacher model and thus share the same prior knowledge about input samples. Therefore, optimizing joint scores encourages models to preserve prior knowledge when adapting to new classes.

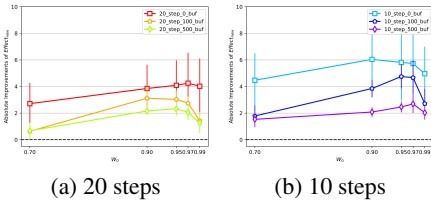

(a) 20 steps       (b) 10 steps

Figure 6: The absolute improvements of $Effect_{new}$.

Table 4: The hyper-parameter analysis of $Effect_{old}$.

| Buffer Size | Method | 20 step | | 10 step | |
|---|---|---|---|---|---|
| | | AverACC (↑) | Δ AverACC | AverACC (↑) | Δ AverACC |
| 0 | BaCE w/o $Effect_{old}$ ($\alpha = 0$) | 19.53 | / | 29.43 | / |
| | BaCE ($\alpha = 1$) | 23.54 | +4.01 | 37.22 | +7.79 |
| | BaCE ($\alpha = 5$,Ours) | **35.36** | **+15.83** | **51.84** | **+22.41** |
| 100 | BaCE w/o $Effect_{old}$ ($\alpha = 0$) | 57.15 | / | 64.28 | / |
| | BaCE ($\alpha = 0.01$) | 57.68 | +0.53 | 65.21 | +0.93 |
| | BaCE ($\alpha = 0.1$) | 64.13 | +6.98 | 67.73 | +3.45 |
| | BaCE ($\alpha = 1$,Ours) | **65.88** | **+8.73** | **74.81** | **+10.53** |
| 500 | BaCE w/o $Effect_{old}$ ($\alpha = 0$) | 78.54 | / | 78.60 | / |
| | BaCE ($\alpha = 0.01$) | 78.71 | +0.17 | 78.96 | +0.36 |
| | BaCE ($\alpha = 0.1$) | 81.48 | +2.94 | 79.64 | +1.04 |
| | BaCE ($\alpha = 1$,Ours) | **82.46** | **+3.92** | **84.59** | **+5.99** |

**Hyper-parameter Analysis.** The hyper-parameters analysis is conducted on split CIFAR-100 with ViT-B/16. Fig. 6 shows the difference between BaCE and BaCE w/o $Effect_{new}$) when different $W_0$ is selected. It indicates that the model has robust performance when $W_0 = 0.95$. Table 4 indicates that $Effect_{old}$ (Eq.1) brings considerable improvement based on DER++. We set $\alpha = 1$ when the buffer is available and $\alpha = 5$ when it is unavailable. The results of other hyper-parameters are demonstrated in the Appendix D.1.

# 6   CONCLUSION

In this research, we start from a contradictory phenomenon in recent studies and discover that classifiers forget much faster than PTMs. To find out the cause, we conduct a pilot study based on linear probing and reveal that the confrontation phenomenon leads to the forgetting in classifiers. To this end, we propose BaCE to mitigate the confrontation phenomenon by balancing the causal effects between new and old data when adapting to each class. Different from prior CIL methods, BaCE tackles the confrontation phenomenon at the root by promoting models to learn new and old data jointly with the balanced mutual causal effects. Finally, we verify the effectiveness of BaCE through extensive experiments on both visual and NLP tasks.

There are two main limitations of this research. The proposed method does not fully address the confrontation phenomenon, and the disparity with upper limit performance is still large when the buffer size is small. Furthermore, the computation cost during training may increase by 2 to 5 times.

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

APPENDIX

## A  RELATED WORK

### A.1  CLASS INCREMENTAL LEARNING

Apart from the related work in the main paper, there are some other directions in CIL. Wortsman et al. (2020) finds subnetworks for each task during training and infers the task using gradient-based optimization at inference. Since the model parameters are not updated in the training, this method does not suffer from catastrophic forgetting. However, finding subnetworks prohibits the potential forward and backward knowledge transfer in CIL. In our experiments, we find that freezing the parameters of the encoder and the old classifier does not lead to satisfactory performance. Kim et al. (2022b) decomposes the CIL problem into within-task prediction and task-id prediction and proposes using out-of-distribution detection techniques for inferring task id. Kim et al. (2022a) proposes to train a multi-head classifier with an out-of-distribution class for each head and utilizes the adapter to prevent interference between the parameters of different tasks. Kim & Han (2023)

proposes two techniques for alleviating the stability-plasticity dilemma. The first technique improves an architecture-based method DER Yan et al. (2021). The second technique freezes the parameters of the encoder and the old classifier, which is considered as a baseline in our experiments. Arani et al. (2022) improves the replay buffer system using the complementary learning system theory. Unlike Arani et al. (2022), BaCE is built based on the theory of causal inference and does not rely on the replay buffer. The introduction of the CIL methods considered in our experiments of continual image classification is in the Appendix D.1.

## A.2 Incremental Learning with PTMs

Apart from the related work in the main paper, there are some other task-specific CIL methods in natural language processing using PTMs, such as BERT. Wang et al. (2020) improves MBPA++ (de Masson D'Autume et al., 2019) and proposes a meta-lifelong framework for text classification and question answering. Sun et al. (2020) utilizes the pretrained knowledge in gpt2 (Radford et al., 2019) and generates pseudo-samples of old tasks when learning new tasks. Xia et al. (2022) proposes a two-stage framework Learn-and-Review for continual NER. It utilizes knowledge distillation and generates pseudo-samples of old entity types when learning new tasks. Zhang et al. (2023) improves the knowledge distillation to take advantage of existing translation models. Xia et al. (2023) proposes to split the last layer into previous and current classifiers to mitigate the classifier bias and representation bias for continual relation extraction. The introduction of the CIL methods considered in our experiments of continual text classification and continual image classification is in the Appendix D.1.

## A.3 Probing Study in Incremental Learning

Most previous studies measured catastrophic forgetting by evaluating performance drops on old tasks. Probing is another useful technique to evaluate the representation ability of a model. However, there is little understanding about the probing performance in Incremental learning. Davari et al. (2022) uses linear probing to show that the representations still suffer from significant drift due to parameter updates even when performance on previously learned tasks can be preserved. Different from Davari et al. (2022), our study utilizes a probing study to show that classifiers forget much faster than PTMs. Wu et al. (2021) conducts layer-wise probing studies on BERT and shows that catastrophic forgetting happens in the top and middle layers. They also indicate that although BERT still maintains a high representative ability at the last incremental step, the classifier has already lost the ability to classify previously learned classes. Unlike Wu et al. (2021), this work considers a more comprehensive scenario in the probing study, including three settings (SEQ, REPLAY, and MTL) and two backbones (ViT-B/16 and ResNet-18). Tao et al. (2023) used linear probing to show that BERT is inherently resilient to catastrophic forgetting even without buffer data in Task-Incremental Learning. Our work focus utilizes probing study to investigate Class-Incremental Learning, which is a more challenging and complicated scenario in incremental learning. Chen et al. (2023) conducted linear probing on k-shot samples from the next task to show a strong correlation between retaining past information and learning efficiency on new tasks.

## A.4 Class Imbalanced Problem in CIL

The class imbalanced problem (Japkowicz & Stephen, 2002; He & Garcia, 2009) between old and new classes is a long-standing problem in CIL, and a lot of studies have attempted to alleviate this problem. For example, LUCIR (Hou et al., 2019) proposes using a cosine classifier to avoid the imbalanced magnitudes between new and old predictions. IL2M (Belouadah & Popescu, 2019) introduces an additional memory for storing the statistics of old tasks obtained when they were initially learned. BiC (Wu et al., 2019) addresses the data imbalance between the old and new classes by fine-tuning classifiers on balanced data. The confrontation phenomenon described in this study is closely related to the class imbalanced problem. Both of them imply that new and old tasks should be learned in a balanced way. The difference is that the confrontation phenomenon describes the whole learning process at the feature level, while the class imbalanced problem only describes the prediction bias at inference time. Therefore, the confrontation phenomenon motivates us to learn new and old tasks with balanced effects from both new and old data, while the class imbalanced problem motivates previous works to design models that give balanced predictions.

### A.5 Causal Inference in CV and NLP

Causal inference (Glymour et al., 2016; Pearl, 2009) has been recently introduced to various visual and NLP tasks such as image classification (Hu et al., 2021), long-tailed classification (Tang et al., 2020; Nan et al., 2021), distantly supervised named entity recognition (Zhang et al., 2021), neural dialogue generation (Zhu et al., 2020), continual named entity recognition (Zheng et al., 2022), and fine-tuning (Zheng et al., 2023). Our idea stems from the causal view on forgetting in Hu et al. (2021), and the proposed BaCE further seeks a balance between the causal effects of new and old data.

### A.6 Continual Causal Discovery

Causality theory (Pearl, 2009) provides language, algorithms, and tools to discover and infer cause-and-effect relationships from any collection of observational/experimental data based on a partial understanding of a complex system. Despite causality having taken huge strides in recent years, few studies consider the continual learning setting in causal discovery (Mundt et al., 2023). Javed et al. (2020); Chu et al. (2020); Gong et al. (2023) focused on learning causal structures from a data stream. Unlike them, our research focuses on class-incremental learning instead of finding causal structures behind data.

## B  Probing and Tracking Studies

### B.1  Training Settings.

The training settings are the same as in the experiments of Continual Image Classification. When using ViT-B/16 as the backbone, we train the model for 20 epochs and probe the model every four epochs. Specifically, we randomly initialize the classifier and train five epochs with a learning rate of 0.1 on all data of the tasks learned so far. When using ResNet-18 as the backbone, we train the model for 120 epochs and probe the model every 30 epochs. The classifier is re-trained for 15 epochs with a learning rate of 0.1.

We use a linear classifier with cosine normalization (Hou et al., 2019) (i.e., cosine classifiers). The cosine classifier has a weight matrix without bias. For example, when learning the second task, the shape of the weight matrix is $10 \times 768$. Specifically, 10 is the number of categories learned so far, and 768 is the hidden dimension of the encoder. After training the second task, we expand the weight matrix to $15 \times 768$ and randomly initialize the new parameters. Since the logits are computed as the cosine similarity between features and the row vectors in the weight matrix, each row vector can be regarded as the representation learned by the classifier. In this paper, we call them class embeddings for clarity.

### B.2  Definition of Probing and Original Performance.

We explain how "Probing Accuracy", "Original Accuracy", "Average Probing Accuracy", and "Average Original Accuracy" are computed in the probing study. Recall that a model has two components: the encoder and the classifier. During the CIL training, the encoder and the classifier are not frozen. For example, in Fig. 8, we train a model normally from task 1 to task 20 and save model checkpoints every 30 epochs. Since we train the model for 120 epochs in each incremental task, we obtain $20 \times (120/30 + 1) = 100$ model checkpoints. For each checkpoint, we can compute its "Probing Accuracy", "Original Accuracy", "Average Probing Accuracy", and "Average Original Accuracy" as follows:

- To obtain the "Original Accuracy" of one task, we evaluate the model checkpoint on that task. For example, to obtain the "Original Accuracy" of task 15, we evaluate the test accuracy of task 15.

- To obtain "Average Original Accuracy", we take the average of the "Original Accuracy"s of all tasks that the model checkpoint has learned. For example, to obtain the "Average Original Accuracy" of a model checkpoint in task 15, we take the average of the "Original Accuracy" s of tasks 1,2,*cdots*,15.

- To obtain the "Probing Accuracy" of one task, we first re-initialize the classifier, freeze the encoder, and train the re-initialized classifier on all seen tasks' data. After training the classifier, we obtain a modified model checkpoint and evaluate its test accuracy on that task. We note that the modified model checkpoint will NOT be used for the subsequent CIL training.

- To obtain "Average Probing Accuracy", take the average of the "Probing Accuracy" s of all tasks that the model checkpoint has learned.

In the end, we measure the "Probing Accuracy", "Original Accuracy", "Average Probing Accuracy", and "Average Original Accuracy" of all 100 model checkpoints and obtain Fig. 8.

In summary, "Origin Accuracy" and "Average Original Accuracy" refer to CIL models' accuracy and average accuracy without additional training. "Probing Accuracy" and "Average Probing Accuracy" measure the ideal performance when the classifier does not forget.

## B.3 ADDITIONAL RESULTS ON VIT.

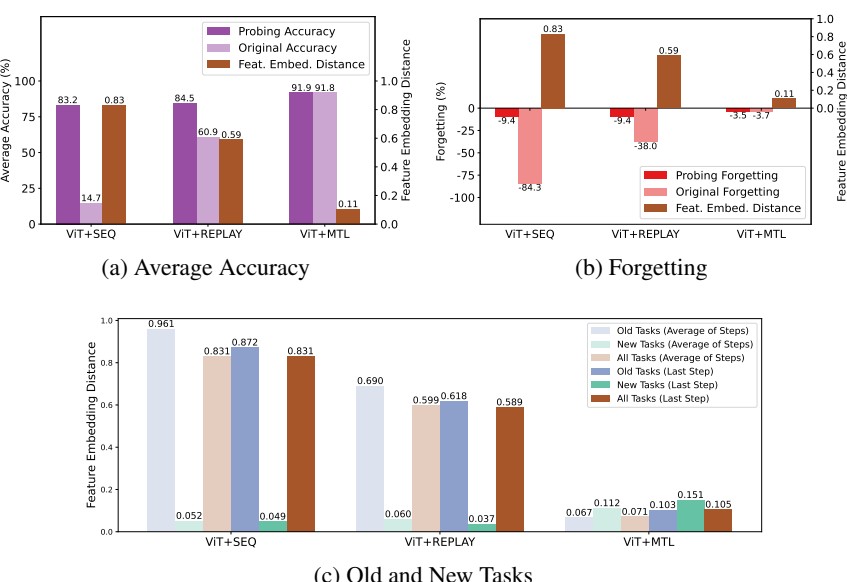

(a) Average Accuracy  (b) Forgetting

(c) Old and New Tasks

Figure 7: The relationship between feature-embedding distance and (a) average accuracy and (b) forgetting when using ViT-B/16 as the backbone. (c) shows the feature-embedding distance of old and new tasks. "Average of Steps" means the distance is averaged over all incremental steps;"Last Step" means the distance is calculated at the last incremental step.

Fig. 7a shows that when more replay data is available, the feature-embedding distance decreases, and the gap between probing and original performance is narrowed. It indicates that minimizing the feature-embedding distance may close the performance gap. Fig. 7b shows a similar trend from the perspective of forgetting. Fig. 7c shows that the feature-embedding distance of new tasks averaged over all incremental steps increases slightly while that of old tasks dramatically decreases when training with more replay data. It indicates that the new and the old tasks are trained in a confrontational manner. The confrontation phenomenon will not hurt the performance when all new and old data are trained jointly,i.e., the MTL setting. However, when only limited old data is stored, the confrontational effects may hinder models from learning the optimal representations of both new and old data.

## B.4 ADDITIONAL RESULTS ON RESNET-18.

The probing and tracking study results with ResNet-18 are provided in Fig. 8 and Fig. 9, respectively. Fig. 8 indicates that ResNet-18 heavily relies on rehearsal data to preserve knowledge. Besides,

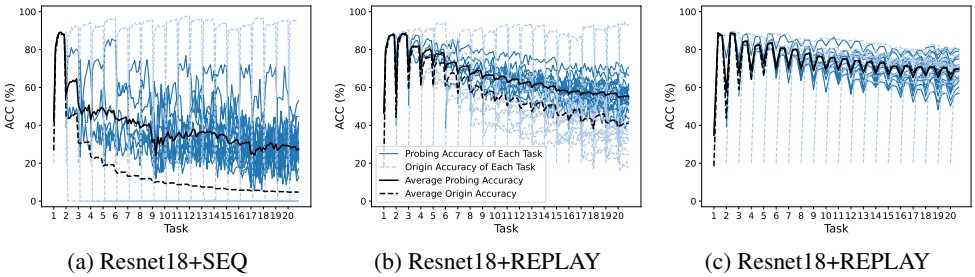

(a) Resnet18+SEQ      (b) Resnet18+REPLAY      (c) Resnet18+REPLAY

Figure 8: The probing study on the 20-step split CIFAR-100 (Krizhevsky et al., 2009). We use randomly initialized Resnet-18 (He et al., 2016) as the backbone. The buffer size is 2000 in REPLAY. The blue curve represents the accuracy on each task, and the black curve represents the average accuracy on all tasks learned so far. The solid and the dotted line represent the probing and original accuracy, respectively.

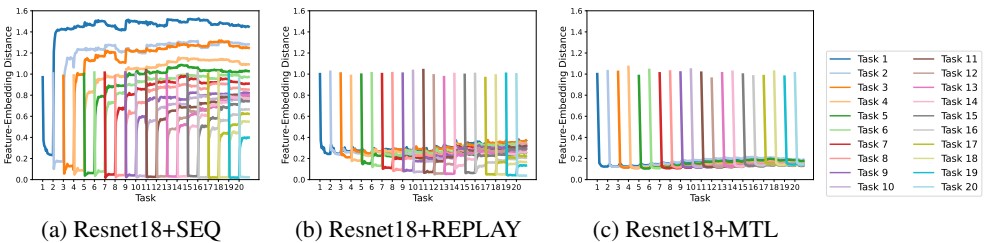

(a) Resnet18+SEQ      (b) Resnet18+REPLAY      (c) Resnet18+MTL

Figure 9: The evolution of feature-embedding distance in CIL. The backbone is ResNet-18, and the dataset is the 20-step split CIFAR-100. Each colour represents the average feature-embedding distance of classes from an incremental task.

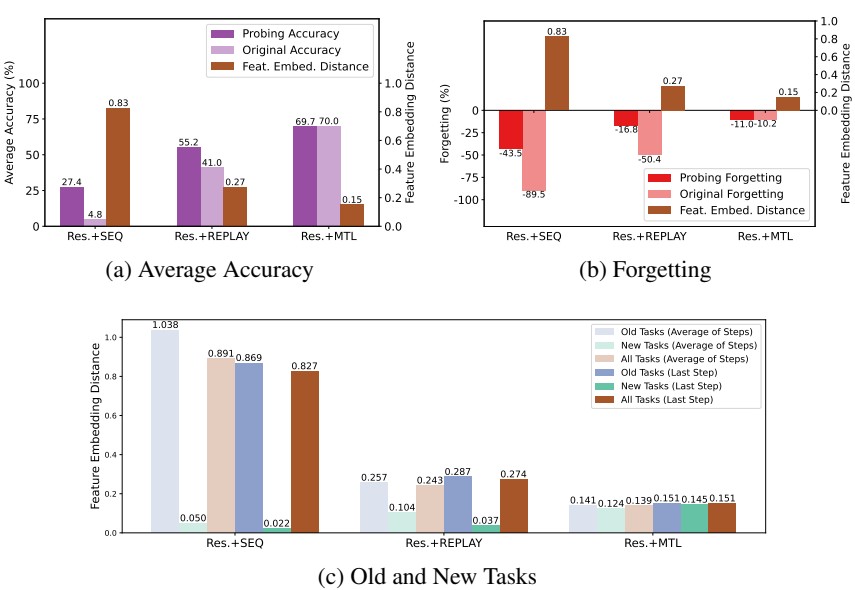

(a) Average Accuracy          (b) Forgetting

(c) Old and New Tasks

Figure 10: The relationship between feature-embedding distance and (a) average accuracy and (b) forgetting when using ResNet-18 as the backbone. (c) shows the feature-embedding distance of old and new tasks. "Average of Steps" means the distance is averaged over all incremental steps; "Last Step" means the distance is calculated at the last incremental step.

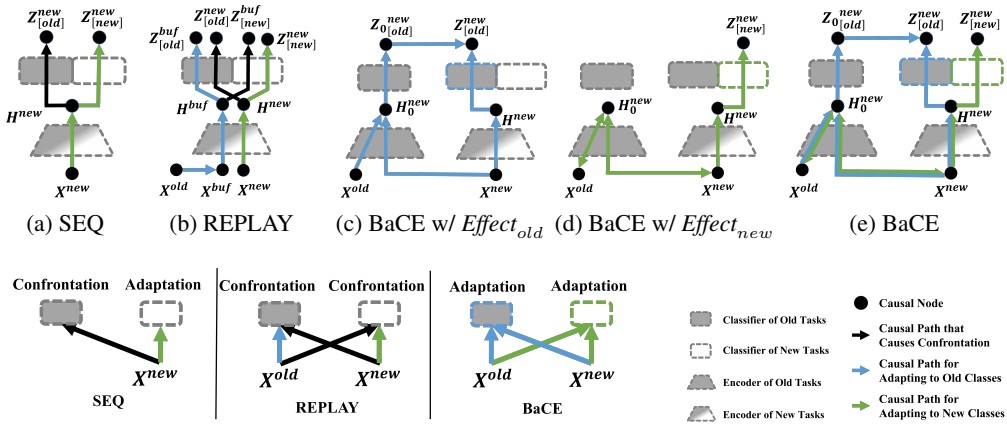

(a) SEQ    (b) REPLAY    (c) BaCE w/ $Effect_{old}$   (d) BaCE w/ $Effect_{new}$    (e) BaCE

(f) Comparison Between SEQ, REPLAY and BaCE

Figure 11: The Causal Graph of SEQ, REPLAY, and BaCE (Ours) with $Effect_{old}$ and $Effect_{new}$ in each CIL step. The directed edges represent the causal effects between variables in the forward propagation. The blue and green paths represent the causal effects of adapting to old and new classes, respectively, when taking backpropagation into consideration. Besides, the black paths represent the causal effects that cause confrontation.

the gap between probing and original performance is smaller than that of pre-trained ViT when no buffer is available. Similar to the trend in ViT, more replay data closes the gap to the upper limit performance. Fig. 9 shows the confrontation phenomenon also exists in ResNet-18.

Fig. 10a and 10b show the relationship between feature-embedding distance and average accuracy and forgetting, respectively. Fig. 10c compares the feature embedding distance between new and old tasks, showing similar trends as in ViT.

## B.5 SUMMARY

We summarize the findings and insights from the probing and tracking studies:

- The catastrophic forgetting primarily happens in the classifier when using pre-trained ViT-B/16 for CIL.
- The performance gap in pre-trained ViT-B/16 is larger than randomly-initialized ResNet-18.
- The confrontation phenomenon accelerates the forgetting in the classifier.
- The confrontation phenomenon is severe when limited replay data is available, and replaying more old data helps alleviate the confrontation phenomenon.

## C  FURTHER EXPLANATION OF BACE

### C.1  PROOF OF BALANCED CAUSAL EFFECTS FOR $Effect_{old}$

This subsection proves that optimizing $Effect_{old}$ builds causal paths from $X^{old}$ and $X^{new}$ to $Z^{new}_{[old]}$ as shown in Fig. 11c. We define $Effect_{old}$ as:

$$Effect_{old} = \mathbb{E}_{(x,y)\sim\mathcal{D}_t}(-\alpha\mathcal{L}_{KL}(S^{new}_{[old]}(x), S0^{new}_{[old]}(x)) \tag{5}$$

$\mathcal{L}_{KL}(\cdot,\cdot)$ is the Kullback-Leibler Divergence. $S^{new}_{[old]}$ and $S0^{new}_{[old]}$ are the scores of old classes output by $f^t$ and $f^{t-1}$. $\alpha$ is the scaling hyper-parameter. And the rehearsal data further enhances $Effect_{old}$ as follows:

$$Effect_{old-replay} = Effect_{old} + \mathbb{E}_{(x,y)\sim\mathcal{M}}(-\mathcal{L}_{CE}(x,y) - ||Z^{buf}_{[old]}(x) - Z0^{buf}_{[old]}(x)||^2_2) \tag{6}$$

$\mathcal{L}_{CE}(\cdot, \cdot)$ is the cross-entropy loss. $Z_{[old]}^{buf}$ and $Z_{0[old]}^{buf}$ are the logits of old classes output by $f^t$ and $f^{t-1}$. $|| \cdot ||_2$ is the Euclidean distance.

Then, we prove Eq. 1 encourages models to learn old classes with causal effects from both new and old data. We defined $Effect_{old}$ as the difference between the logits when $X^{old}$ exists or not:

$$Effect_{old} = \mathbb{P}(Z_{[old]}^{new}|do(X^{old} = x^{old})) - \mathbb{P}(Z_{[old]}^{new}|do(X^{old} = 0)) \qquad (7)$$

Since $X^{old}$ has no parent nodes, we simplify it as follows:

$$Effect_{old} = \mathbb{P}(Z_{[old]}^{new}|X^{old} = x^{old}) - \mathbb{P}(Z_{[old]}^{new}|X^{old} = 0) \qquad (8)$$

Then, we expand the above equation and obtain:

$$Effect_{old} = \mathbb{P}(Z_{[old]}^{new}|Z_{0[old]}^{new}, X^{old} = x^{old})\mathbb{P}(Z_{0[old]}^{new}|X^{old} = x^{old})$$
$$- \mathbb{P}(Z_{[old]}^{new}|Z_{0[old]}^{new}, X^{old} = 0)\mathbb{P}(Z_{0[old]}^{new}|X^{old} = 0) \qquad (9)$$
$$= \mathbb{P}(Z_{[old]}^{new}|Z_{0[old]}^{new})\mathbb{P}(Z_{0[old]}^{new}|X^{old} = x^{old}) - \mathbb{P}(Z_{[old]}^{new}|Z_{0[old]}^{new})\mathbb{P}(Z_{0[old]}^{new}|X^{old} = 0) \qquad (10)$$
$$= \mathbb{P}(Z_{[old]}^{new}|Z_{0[old]}^{new})(\mathbb{P}(Z_{0[old]}^{new}|X^{old} = x^{old}) - \mathbb{P}(Z_{0[old]}^{new}|X^{old} = 0)) \neq 0 \qquad (11)$$

The Eq. 10 holds because $Z_{0[old]}^{new}$ is the only mediator (Pearl, 2009) from $X^{old}$ to $Z_{[old]}^{new}$. The Eq. 11 holds because $Z_{0[old]}^{new}$ is regularized by $Z_{[old]}^{new}$, and $\mathbb{P}(Z_{0[old]}^{new}|X^{old} = x^{old}) \neq \mathbb{P}(Z_{0[old]}^{new}|X^{old} = 0)$.

Furthermore, $Z_{[old]}^{new}$ is obtained from $X^{new}$ and thus $X^{new}$ has causal effects on $Z_{[old]}^{new}$. Therefore, we prove that both $X^{old}$ and $X^{new}$ have causal effects on $Z_{[old]}^{new}$ when optimizing $Effect_{old}$.

## C.2 The connection between $Effect_{old}$ and existing works

The objective of $Effect_{old}$ is the same as the knowledge distillation term in LWF (Li & Hoiem, 2017). The latter term of $Effect_{old-replay}$ is the same as the regularization term in DER++ (Buzzega et al., 2020). In the experiment, we observe that it is beneficial to encourage teacher and student response similarly to data points of the current task when data replay, which is overlooked by DER++. From the causal perspective, $Effect_{old}$ encourages models to preserve old knowledge with the causal effect of new data. Therefore, adapting to old classes has a less negative impact on adapting to new classes, i.e., the confrontation phenomenon is mitigated.

## C.3 Proof of Balanced Causal Effects for $Effect_{new}$

This subsection proves that optimizing $Effect_{new}$ builds causal paths from $X^{old}$ and $X^{new}$ to $Z_{[new]}^{new}$ as shown in Fig. 11d. We denote the prediction score over categories as $S$, obtained from logits through the softmax function. Firstly, $Effect_{new}$ can be defined as the difference between the score prediction of $X^{new}$ when $X^{old}$ exists or not.

$$Effect_{new} = \mathbb{P}(S^{new}|H_0^{new}, do(X^{old} = x^{old})) - \mathbb{P}(S^{new}|H_0^{new}, do(X^{old} = 0)) \qquad (12)$$

$do(\cdot)$ is the *do-operation*, which represents assigning a certain value to a variable without considering its parent nodes. $\mathbb{P}(S^{new}|H_0^{new}, do(X^{old} = x^{old}))$ is the score prediction when $f^{t-1}$ is trained on $x^{old}$. $\mathbb{P}(S^{new}|H_0^{new}, do(X^{old} = 0))$ is the score prediction when $f^{t-1}$ is trained without old data, i.e., $f^{t-1}$ is randomly-initialized. We note that $Effect_{new}$ is defined as the difference between scores instead of logits because scores may be invariant when logits change. In Eq. 12, the prediction score $S^{new}$ conditions on the collider $H_0^{new}$. In this case, $X^{old}$ has causal effects on $X^{new}$ through the collider $H_0^{new}$.

Then, we re-write $Effect_{new}$ as the sum of the causal effect on each sample's prediction:

$$Effect_{new} = \sum_i^N Effect_{new}^{(i)} \qquad (13)$$

By introducing the definition in Eq. 12, we have:

$$Effect_{new}^{(i)} = \mathbb{P}(S^{new(i)}|H_0^{new} = h_0^{(i)}, do(X^{old} = x^{old})) - \mathbb{P}(S^{new(i)}|H_0^{new} = h_0^{(i)}, do(X^{old} = 0)) \qquad (14)$$
$$= \mathbb{P}(S^{new(i)}|H_0^{new} = h_0^{(i)}, X^{old} = x^{old}) - \mathbb{P}(S^{new(i)}|H_0^{new} = h_0^{(i)}, X^{old} = 0) \qquad (15)$$

$S^{new(i)}$ is the score prediction of the $i$-th sample $x^{(i)}$. $h_0^{(i)}$ is the feature of $x^{(i)}$ extracted by the encoder of $f^{t-1}$(denoted as $f_{enc}^{t-1}$), i.e., $h_0^{(i)} = f_{enc}^{t-1}(x^{(i)})$. $N$ is the number of samples in $X^{new}$. Eq. 14 defines the causal effect on each sample ($Effect_{new}^{(i)}$) as the difference between the score prediction of $x^{(i)}$ when $f^{t-1}$ is trained on $x^{old}$ or not. Eq. 15 holds because $X^{old}$ has no parent nodes.

And Then, $Effect_{new}^{(i)}$ is estimated as follows:

$$Effect_{new}^{(i)}$$

$$= \sum_{k=1}^{N} (\mathbb{P}(S^{new(i)}|X^{new} = x^{(k)}, H_0^{new} = h_0^{(i)})(\mathbb{P}(X^{new} = x^{(k)}|H_0^{new} = h_0^{(i)}, X^{old} = x^{old})$$

$$\tag{16}$$

$$- \mathbb{P}(X^{new} = x^{(k)}|H_0^{new} = h_0^{(i)}, X^{old} = 0))$$

$$= \sum_{k=1}^{N} (\mathbb{P}(S^{new(i)}|X^{new} = x^{(k)})(\mathbb{P}(X^{new} = x^{(k)}|H_0^{new} = h_0^{(i)}, X^{old} = x^{old}) \tag{17}$$

$$- \mathbb{P}(X^{new} = x^{(k)}|H_0^{new} = h_0^{(i)}, X^{old} = 0))$$

$$\approx \sum_{k=1}^{N} \mathbb{P}(S^{new(i)}|X^{new} = x^{(k)}) \underbrace{\mathbb{P}(X^{new} = x^{(k)}|H_0^{new} = h_0^{(i)}, X^{old} = x^{old})}_{W_{i,k}} \tag{18}$$

$$\approx \sum_{k=1}^{K} \mathbb{P}(S^{new(i)}|X^{new} = x^{(k)})W_{i,k} \tag{19}$$

Eq. 16 is obtained by applying the Bayes Rule to Eq. 15. Eq. 17 holds since $X^{new}$ is the only mediator (Pearl, 2009) from $X^{old}$ to $S^{new(i)}$. Eq. 18 approximates $\mathbb{P}(X^{new} = x^{(k)}|H_0^{new} = h_0^{(i)}, X^{old} = 0)$ as zero because the likelihood is small when $f^{t-1}$ is randomly initialized. We further expand the latter term in Eq. 18 using the Bayes Rule as follows:

$$\mathbb{P}(X^{new} = x^{(k)}|H_0^{new} = h_0^{(i)}, X^{old} = x^{old}) =$$

$$\frac{\mathbb{P}(H_0^{new} = h_0^{(i)}|X^{new} = x^{(k)}, X^{old} = x^{old})\mathbb{P}(X^{new} = x^{(k)}|X^{old} = x^{old})}{\mathbb{P}(H_0^{new} = h_0^{(i)}|X^{old} = x^{old})} \tag{20}$$

In Eq. 20, $\mathbb{P}(H_0^{new} = h_0^{(i)}|X^{old} = x^{old})$ and $\mathbb{P}(X^{new} = x^{(k)}|X^{old} = x^{old})$ are intractable and we regard them as constants. Then, $\mathbb{P}(X^{new} = x^{(k)}|H_0^{new} = h_0^{(i)}, X^{old} = x^{old})$ mainly depends on the likelihood term $\mathbb{P}(H_0^{new} = h_0^{(i)}|X^{new} = x^{(k)}, X^{old} = x^{old})$, which represents how likely the hidden feature is $h_0^{(i)}$ when the input sample is $x^{(k)}$. Obviously, the likelihood is the largest when $k = i$ and becomes smaller when the hidden feature of $x^{(k)}$ becomes farther away from $h_0^{(i)}$. Recall that $h_0^{(i)}$ is the hidden feature of $x^{(i)}$ extracted by $f_{enc}^{t-1}$. The latter term in Eq. 18 can be regarded as the scaling factor, which is determined by the distance of $x^{(i)}$ and $x^{(k)}$ in the feature space of $f_{enc}^{t-1}$. Considering estimating Eq. 18 on all training samples is prohibitive due to time and space, we truncate top-K samples and obtain Eq. 19. In summary, Eq. 19 computes $Effect_{new}^{(i)}$ as the weighted sum of $\mathbb{P}(S^{new(i)}|X^{new} = x^{(k)})$ on the K-Nearest-Neighbours of $x^{(i)}$. And the weight $W_{i,k}$ is larger when the distance between $x^{(k)}$ and $x^{(i)}$ in the feature space of $f_{enc}^{t-1}$ is smaller. Noteworthily, when $k = i$, $\mathbb{P}(S^{new(i)} = y^{(i)}|X^{new} = x^{(i)})$[3] is exactly the likelihood we expected to maximize. Therefore, maximizing $Effect_{new}$ amounts to minimizing the classification loss of each sample, except that the score is the joint score estimated by itself and its neighbors.

On these grounds, $Effect_{new}^{(i)}$ is estimated as follows:

$$Effect_{new}^{(i)} = -\mathcal{L}_{CE}(W_0 s^{(i)} + \sum_{k \in [1, \cdots, K]} W_{i,k} s^{(i,k)}, y^{(i)}), \tag{21}$$

---

[3] $y^{(i)}$ is regarded as a one-hot distribution here.

where $s^{(i)}$ and $s^{(i,k)}$ are the score prediction of $x^{(i)}$ and the $k$-th neighbours of $x^{(i)}$. $W_0$ is the weight of the input sample. $W_{i,k}$ is the weight of neighbors and it is defined as follows:

$$W_{i,k} = \frac{(1 - W_0)/||H_0(x^{(i,k)}) - H_0(x^{(i)})||_2}{\sum_{k' \in [1,\cdots,K]} 1/||H_0(x^{(i,k')}) - H_0(x^{(i)})||_2}, \tag{22}$$

where $x^{(i,k)}$ is the $k$-th neighbour of $x^{(i)}$; $H_0(x^{(i,k)})$ and $H_0(x^{(i)})$ are the feature of $x^{(i,k)}$ and $x^{(i)}$ extracted by $f_{enc}^{t-1}$; and the denominator is a normalized term. In Eq. 4, the weights of neighbors are defined as the normalized reciprocal of Euclidean distance to input samples in the feature space of $f_{enc}^{t-1}$. When $W_0 = 1$, $Effect_{new}^{(i)}$ degenerates to $\mathcal{L}_{CE}(s^{(i)}, y^{(i)})$.

In summary, $Effect_{new}$ is estimated as follows:

$$Effect_{new} = -\mathbb{E}_{(x,y) \sim \mathcal{D}_t} \mathcal{L}_{CE}(\sum_{\tilde{x} \in \{x\} \cup \mathcal{N}(x)} W(\tilde{x}, x) S(\tilde{x}), y), \tag{23}$$

where $\sum_{\tilde{x} \in x \cup \mathcal{N}(x)} W(\tilde{x}, x) = 1$; $\mathcal{N}$ is the set of K-Nearest-Neighbors (KNNs) in the feature space of $f^{t-1}$; $S(\tilde{x})$ is the score prediction of $\tilde{x}$; $W(\tilde{x}, x)$ is the weight of $S(\tilde{x})$ and it is defined as follows:

$$W(\tilde{x}, x) = \begin{cases} W_0, & \text{when} \quad \tilde{x} = x; \\ \frac{(1-W_0)/||H_0(\tilde{x}) - H_0(x)||_2}{\sum_{\tilde{x}' \in \mathcal{N}(x)} 1/||H_0(\tilde{x}') - H_0(x)||_2}, & \text{otherwise.} \end{cases} \tag{24}$$

In Fig. 11d, the causal path from $X^{old}$ to $Z_{[new]}^{new}$ is built by optimizing Eq. 3. Although $S^{new}$ is determined by both $Z_{[new]}^{new}$ and $Z_{[old]}^{new}$, only the causal effect between $S^{new}$ and $Z_{[new]}^{new}$ helps models adapt to new classes.

Furthermore, $Z_{[new]}^{new}$ is obtained from $X_{new}$, and thus $X_{new}$ has causal effects on $Z_{[new]}^{new}$. Therefore, we prove that both $X_{old}$ and $X_{new}$ have causal effects on $Z_{[new]}^{new}$ when optimizing $Effect_{new}$.

### C.4 The connection between $Effect_{new}$ and existing works

$Effect_{new}$ is inspired by DDE (Hu et al., 2021). Different from the collider effect in DDE, $Effect_{new}$ further estimates the weight of neighbors based on the Euclidean distance to input samples and thus fundamentally works better than DDE. Furthermore, BaCE updates the neighborhood relationships as well as the teacher model every training epoch, which promotes adaptation to the distribution of new data.

### C.5 The connection between BaCE and existing works

We note that the motivation of BaCE is different from the class imbalanced problem. We provide a detailed explanation in Appendix A.4. Unlike the existing techniques for the class imbalanced problem, BaCE exploits the causal effects of new and old data jointly and allows forward and backward transfer during training on new data. Finally, we summarize the algorithm in Alg. 1.

### C.6 Why BaCE mitigates the confrontation phenomenon?

Recall that in the tracking study, we find that the learning process between new and old data is "confrontational". From the causal perspective, the confrontation phenomenon is caused by the confrontational causal effects between new and old tasks. Recall that in the tracking study of SEQ, the model only adapts to new data. Fig. 2a shows that in this case, (a) the class features and class embeddings in new tasks reach their optimal positions, and (b) the class features and class embeddings in old tasks are pushed away from their optimal positions (also see the illustration in Fig. 3b).

Unlike SEQ, in the setting of REPLAY and MTL, the losses of new and old data are jointly optimized. Intuitively, the effect of optimizing the joint loss can be disentangled into two separate effects, i.e., the effect of optimizing the loss of new data and the effect of optimizing the loss of old data. In this view, jointly optimizing old data's loss and new data's loss leads to two confrontational effects, i.e., the effect of adapting to new data and hindering the learning of old tasks and the effect of adapting to old data and hindering the learning of new tasks. Furthermore, these two confrontational effects

exist even in the MTL setting. Since all old data is available in MTL, the distribution of both new and old data can be modeled properly, and the two confrontational effects are neutralized. However, in the practical setting of CIL, the buffer size is limited (i.e., the REPLAY setting). This means that the rehearsal samples may not reflect the true data distribution of old tasks. In this case, the two confrontational effects are not neutralized, and it may result that the new task is well learned while the old tasks are forgotten.

Additionally, we display a thought experiment to demonstrate our key intuition better. Suppose we start from the MTL setting and gradually reduce the number of the new task's samples to zero. If we conduct a tracking study on this model, we may observe that the feature-embedding distance of the new task increases monotonously. In contrast, the feature-embedding distance of the old tasks decreases monotonously. In this case, the two confrontational effects are not neutralized, resulting in the old task being well-preserved while the new task is not being properly learned.

In summary, the tracking study reveals that optimizing old data loss hinders learning new tasks, and optimizing new data loss hinders learning old tasks potentially. In the REPLAY or MTL setting, the confrontational effect of old data is partially or fully counteracted by the confrontational effect of new data. It implies that both new and old tasks may be better learned in the REPLAY setting if we alleviate the confrontational effect of both new and old data.

Motivated by this, we propose BaCE to "simulate" the new/old data effect when learning old/new tasks. BaCE is built upon the balanced causal graphs for pursuing balanced causal effects when learning all classes. We note that the causal graphs in Fig. 11c, 11d and 11e are independent of replay buffer since $X^{old}$ and $X^{buf}$ are different nodes. Therefore, the proposed BaCE is independent of the replay buffer in principle.

### C.7 What is the remaining gap in fully addressing the confrontation phenomenon?

The experimental result in Table 7 shows that the performance gap is still large when buffer size = 0 or 100, although BaCE largely decreases the feature-embedding distance when buffer size = 500. It indicates that the confrontation phenomenon is still severe when only limited old data is available. A natural question is what is the remaining gap in fully addressing the confrontation phenomenon if the causal relations between old/new data and old/new logits are completed as in BaCE.

To this question, we want to clarify that the implementation of $Effect_{new}$ and $Effect_{old}$ can be various, and different implementations may lead to different performance. By implementation, we mean that optimizing other objectives may achieve the same goal from the causal perspective. For example, if we define $Effect_{old}$ as the Cross-Entropy Loss between the teacher's pseudo label and the student's prediction, the performance will differ, although the causal graphs remain unchanged. Another example is that the improvement of $Effect_{new}$ is different when a different value of hyper-parameter $W_0$ is selected. In summary, the proposed causal graphs Fig. 11c,11d, and 11e can be seen as a road map for resolving the phenomenon of confrontation. We hope our findings motivate future studies to investigate the confrontation phenomenon and further close the gap between original performance and probing performance.

## D Additional Experimental Results

### D.1 Continual Image Classification (Continual IC)

**Training Details.** We use SGD as the optimizer for all methods and backbones. The batch size is set as 128. When using ViT-B/16 as the backbone, we train models on each task with a learning rate of 1e-3. The input image is resized to 224×224 to match the pre-training process of ViT. When using ResNet-18 as the backbone, we train models for 120 epochs with an initial learning rate of 1e-2. We use a multi-step scheduler: the milestone is [50, 90], and $\gamma$ is 0.1. The input image is padded to 32×32. The hidden dimensions of ViT-B/16 and ResNet-18 are 768 and 512, respectively. For each method, We exploit the herding algorithm (Rebuffi et al., 2017) to select old representative samples. The implementation is based on PyTorch (Paszke et al., 2019). All experiments are run on GeForce RTX 3090 GPU. We report the average result on three independent runs. We do not use

---

**Algorithm 1:** Balancing Causal Effects (BaCE)

---

**Input:** $\mathcal{D}_t = \{(x^{(i)}, y^{(i)})\}_{i=1}^N$: the training set of the $t$-th task; $f^{t-1}$: the model trained on previous $t-1$ tasks; $K$: the number of neighbors; $\beta$: the hyper-parameter for controlling the update speed of $f^{t-1}$; $\mathcal{M}$: the rehearsal buffer;

**Output:** $f^t$: the model adapted to $t$ tasks

1 Initialize the new model $f^t \leftarrow f^{t-1}$;
2 **while** *not converge* **do**
3      Compute the $K$ nearest neighbors of each sample and obtain $\mathcal{N}$ ;
4      **for** $(x^{(i)}, y^{(i)})$ *in* $\mathcal{D}_t$ **do**
5          Compute $\textit{Effect}_{new}$ according to Eq. 3;
6          **if** $\mathcal{M}$ *is available* **then**
7              Compute $\textit{Effect}_{old-replay}$ according to Eq. 2;
8              $f^t \leftarrow \arg\max_{f^t} \textit{Effect}_{new} + \textit{Effect}_{old-replay}$;
9          **else**
10              Compute $\textit{Effect}_{old}$ according to Eq. 1;
11              $f^t \leftarrow \arg\max_{f^t} \textit{Effect}_{new} + \textit{Effect}_{old}$;
12          **end**
13      **end**
14      $f^{t-1} \leftarrow \beta f^{t-1} + (1-\beta)f^t$;
15 **end**

---

additional data augmentation except for *RandomHorizontalFlip* and *RandomCrop*. The introduction of the backbone models used in the paper is summarized in Table 6.

Table 5: The introduction of the image classification datasets used in the paper.

| | # classes | # training samples | # test samples | Link |
|---|---|---|---|---|
| OmniBenchmark | 300 | 89,697 | 5,985 | Link |
| Tiny-ImageNet | 200 | 100,000 | 10,000 | Link |
| ObjectNet | 200 | 26,509 | 6,628 | Link |
| ImageNet-R | 200 | 24,000 | 6,000 | Link |
| CIFAR100 | 100 | 50,000 | 10,000 | Link |
| VTAB | 50 | 1,796 | 8,619 | Link |
| SVHN | 10 | 73,257 | 26,032 | Link |
| Fashion-MNIST | 10 | 60,000 | 10,000 | Link |
| CIFAR10 | 10 | 50,000 | 10,000 | Link |
| MNIST | 10 | 50,000 | 10,000 | Link |
| Not-MNIST | 10 | 18,265 | 459 | Link |

Table 6: The introduction of the visual transformers used in the paper.

| | # params | Link |
|---|---|---|
| ViT-B/16 | 86M | Link |
| ViT-B/16-IN21K | 86M | Link |
| ResNet18 | 11M | / |
| DeiT-S/16 | 22M | Link |

**Datasets.** The introduction of the image classification datasets used in the paper is summarized in Table 5.

- CIFAR100 (Krizhevsky et al., 2009): CIFAR100 contains 60000 32×32 RGB images of 100 categories. Each class has 500 training images and 100 testing images. We follow Wang et al.

(2022e); Wu et al. (2019) and split 100 classes evenly into 5, 10, and 20 incremental batches. In our experiment, we learned the 100 classes in ascending order. In split CIFAR100, each incremental task shares some similarities since some classes are from the same superclass. Although CIFAR-100 is a simple image classification dataset, existing studies (Wang et al., 2022e) show that pre-trained models still suffer from catastrophic forgetting when the buffer size is small. CIFAR-100 is used following the MIT license.

- 5-datasets (Ebrahimi et al., 2020): We consider a CIL setting proposed in Ebrahimi et al. (2020). 5-datasets consists of five image classification datasets: CIFAR-10 (Krizhevsky et al., 2009), MNIST (LeCun, 1998), Fashion-MNIST (Xiao et al., 2017), SVHN (Netzer et al., 2011), and notMNIST (Bulatov, 2011). CIFAR-10 dataset consists of 60000 $32 \times 32$ color images in 10 classes, with 6000 images per class. Each category contains 5,000 training instances and 1,000 test instances. MNIST is a handwritten digits dataset that is commonly used for benchmarking machine learning algorithms. It contains 10 classes for each digit, and each class has 5000 training instances and 1000 test instances. The fashion-MNIST dataset is a large, freely available database of fashion images. The dataset contains 70000 $28 \times 28$ grayscale images of fashion products from 10 classes, and each class contains 6000 training instances and 1000 test instances. The Street View House Numbers (SVHN) Dataset is a real-world image dataset collected in natural scenes. It contains 73257 training instances and 26032 test instances. The NotMNIST dataset is a collection of $28 \times 28$ grayscale images of 10 different letters (A-J). It contains 18265 training instances and 459 test instances. In our experiments, we train the model according to the same order as in Wang et al. (2022e): SVHN, MNIST, CIFAR10, NotMNIST, Fashion-MNIST. Although each dataset alone is not hard, the sequential training of them is fairly challenging even with ImageNet pre-trained models (Wang et al., 2022e), since models are susceptible to forgetting when the tasks are diverse (Mehta et al., 2021). MNIST is used following the Creative Commons Attribution-Share Alike 3.0 license. CIFAR-10 and Fashion-MNIST are used following the MIT license. The licensing information is not available for SVHN and notMNIST.

- OminiBenchmark: OminiBenchmark is a large benchmark dataset covering more realms and annotating more images of each realm compared with ImageNet-1k Zhang et al. (2022).

- Tiny-ImageNet: Tiny ImageNet has 200 classes, and each class has 500 training images, 50 validation images, and 50 test images. The images are down-sampled to 64 x 64 pixels.

- ObjectNet: ObjectNet is a large real-world test set for object recognition with control where object backgrounds, rotations, and imaging viewpoints are random. Barbu et al. (2019)

- ImageNet-R: ImageNet-R(endition) contains art, cartoons, deviantart, graffiti, embroidery, graphics, origami, paintings, patterns, plastic objects, plush objects, sculptures, sketches, tattoos, toys, and video game renditions of ImageNet classes. ImageNet-R has renditions of 200 ImageNet classes, resulting in 30,000 images. Hendrycks et al. (2021)

- VTAB: The Visual Task Adaptation Benchmark (VTAB) is a diverse and challenging suite of tasks designed to evaluate general visual representations. VTAB defines a good general visual representation as one that performs well on unseen tasks when trained on limited task-specific data. Zhai et al. (2019)

**Discussion of the Overlap between Pretraining and Downstream Datasets.**

The ViT-B/16 is pretrained on ImageNet-21k, which contains 21841 classes ranging from coarse-grained to fine-grained. For example, an image may belong to a label ['animal'→'domestic animal'→'dog'→'spitz'→'Samoyed']. CIFAR-100 contains 100 classes, such as 'lion' and 'tiger'. Obviously, some classes in CIFAR-100 are close to or the same as those in ImageNet-21k. Since ImageNet-21k contains so many images from coarse-to-fine-grained categories, some other widely-used CIL datasets, such as Tiny-ImageNet, Mini-ImageNet, ImageNet-1000/100, inevitably contain similar or the same classes in ImageNet-21k.

In computer vision, existing studies such as L2P (Wang et al., 2022e), DualPrompt (Wang et al., 2022d), ADA (Ermis et al., 2022) and Ramasesh et al. (2022) use pre-trained ViT-B/16 as the backbone and evaluate it on CIFAR100. Apart from 5-datasets, we still use CIFAR-100 for experiments for two reasons: (1) Previous studies such as L2P show that the pretrained ViT-B/16 still suffer from catastrophic forgetting with limited replay data. For example, REPLAY with $10 \times 100 = 1000$ samples

only achieves 67% in the 5-step CIFAR-100 setting. (2) We can quickly compare our method with previous works under exactly the same settings. Future works can readily reproduce or compare with our method.

In IL of natural language processing (NLP), existing studies such as progressive prompt (Razdaibiedina et al., 2023), CT0 (Scialom et al., 2022), LAMOL (Sun et al., 2020), and MBPA++ (de Masson D'Autume et al., 2019) use strong PTMs such as BERT and T5. The pretraining corpus of BERT contains the BooksCorpus (800M words) and English Wikipedia (2,500M words). BERT has likely seen some sentences of AGNews and DBPedia during pretraining. However, existing studies and our study also show much room for improvement in the CIL performance of BERT with limited replay data.

Although the pretraining dataset may overlap with the downstream dataset, Incremental Learning (IL) with Pre-Trained Models (PTMs) is still an under-explored and challenging research topic. More importantly, exploring how to learn novel classes/tasks/domains incrementally with off-the-shell PTMs is significant in the era of PTMs.

**Baselines.**

- ER (Chaudhry et al., 2019): ER is a simple rehearsal method that stores old examples in memory buffers for later replay. Although it is simple, it is an effective and stable baseline.

- LwF (Li & Hoiem, 2017): LwF is a method that exploits knowledge distillation, where the teacher is the model learned on previous tasks and the student is the new model. We set the trade-off parameter in LwF as 5 for a fair comparison.

- EWC (Kirkpatrick et al., 2017): EWC is a regularization-based method that slows down the updates on important parameters. We set the weight of the regularization term as 5000.

- BiC (Hou et al., 2019): BiC is an exemplar-based method that adds a regularization item based on knowledge distillation. After each CIL step, BiC fine-tunes the linear layer and learns two additional parameters to reduce the bias of the backbone network. We follow the original paper and split the exemplar set as 9:1 for training and validation.

- IL2M Belouadah & Popescu (2019): Compared to exemplar-based methods, IL2M additionally stores old class statistics to reduce the prediction bias towards new classes. We reimplement IL2M according to the officially released code in `https://github.com/EdenBelouadah/class-incremental-learning`.

- iCaRL (Rebuffi et al., 2017): iCaRL proposes a herding algorithm for selecting old representative samples. Besides, iCaRL adopts a nearest mean-of-exemplars classifier for classification. We implement iCaRL on ViT according to the officially released code in `http://www.github.com/srebuffi/iCaRL`.

- LUCIR (Hou et al., 2019): LUCIR is an exemplar-based method. LUCIR proposes some modifications to promote separation in the feature space and generate more coordinated incremental learning classifiers. The initial weight of the distillation loss $\lambda_{base}$ is set to 5, $K$ is set to 2, and $m$ is set to 0.5.

- PODNet (Douillard et al., 2020): Apart from the classification loss of new data, PODNet constrains the output of each intermediate layer and the feature output by the backbone network. We use the same default hyper-parameters as in the officially released code in `github.com/arthurdouillard/incremental_learning.pytorch`.

- DDE (Hu et al., 2021): DDE is based on causal inference, which proposes to extract the causal effect between new and old data and capture the incremental momentum effect of the data flow. We use the same default hyper-parameters as in the official released code in `https://github.com/JoyHuYY1412/DDE_CIL`.

- L2P (Wang et al., 2022e): L2P learns a set of prompts that dynamically instruct models to solve corresponding tasks. The set of prompts is called a prompt pool, which is structured in a key-value shared memory space. However, the prompt pool is shared across different tasks, which plays as an implicit external memory in CIL and is unfair to other CIL algorithms (such as BaCE, LwF, DER++, $\cdots$) that require no additional model components. Therefore, we do not compare with L2P when the buffer size is zero in our experiments.

- CLSER (Arani et al., 2022): CLSER improves the replay buffer system by using the complementary learning system (CLS) theory. We use the same default hyper-parameters as in the officially released code in `https://github.com/NeurAI-Lab/CLS-ER`.

- FOSTER (Wang et al., 2022a): FOSTER is a two-stage architecture-based method. In the first stage, FOSTER dynamically expands new modules to fit the residuals between the target and the output of the original model. In the second stage, FOSTER removes redundant parameters and feature dimensions through an effective distillation strategy to maintain the single backbone model. The boosting epochs, compression epochs, and the epochs of the initial task are set to 5. Other hyper-parameters are the same as in the official released code in `https://github.com/G-U-N/ECCV22-FOSTER`.

- MEMO (Zhou et al., 2023a): MEMO is a simple yet effective architecture-based method. MEMO extends specialized layers based on the shared generalized representations. In our experiments, we regard the topmost transformer layer as the specialized layer and the other transformer layers as the task-agnostic layers. We use the same default hyper-parameters as in the officially released code in `https://github.com/wangkiw/ICLR23-MEMO`.

- BEEF (Wang et al., 2023): BEEF is an architecture-based CIL method based on energy-based theory. BEEF decouples the training of independent modules while achieving bi-directional compatibility among modules. The expansion epoch number and the fusion epoch number are set to 3, and the epoch number of the initial task is set to 5. Other hyper-parameters are the same as in the official released code in `https://github.com/G-U-N/ICLR23-BEEF`.

Table 7: The comparison with competitive baselines on split CIFAR-100. All methods use pre-trained ViT-B/16 as the backbone. OOM: Out of GPU memory.

| Buffer Size | Method | 20 step | | | 10 steps | | |
|---|---|---|---|---|---|---|---|
| | | AverACC (↑) | FGT (↓) | FWT (↑) | AverACC (↑) | FGT (↓) | FWT (↑) |
| | SEQ | 14.74 (1.73) | 88.21 (1.21) | 16.18 (2.01) | 24.09 (1.28) | 80.85 (1.79) | 30.83 (3.20) |
| | LwF (Li & Hoiem, 2017) | 30.02 (0.82) | 70.43 (1.94) | 33.81 (1.63) | 45.88 (0.76) | 51.93 (2.30) | 52.96 (1.69) |
| | EWC (Kirkpatrick et al., 2017) | 26.92 (0.77) | 73.86 (1.30) | 27.6 (2.39) | 29.28 (1.02) | 75.26 (2.77) | 32.56 (3.28) |
| 0 | BaCE w/o $Effect_{new}$&$Effect_{old}$ | 15.83 (2.88) | 87.32 (2.15) | 18.39 (2.19) | 23.84 (1.12) | 81.58 (3.52) | 28.16 (1.94) |
| | BaCE w/o $Effect_{new}$ | 31.27 (1.23) | 69.63 (1.63) | 35.35 (1.43) | 46.03 (0.72) | 50.72 (2.03) | 53.70 (1.03) |
| | BaCE w/o $Effect_{old}$ | 19.53 (1.68) | 82.59 (0.85) | 22.94 (1.84) | 29.43 (1.03) | 72.22 (3.39) | 34.97 (2.81) |
| | BaCE (Ours) | **35.36 (0.94)** | **63.20 (1.38)** | **41.96 (2.11)** | **51.84 (1.44)** | **32.99 (2.93)** | **55.24 (3.55)** |
| | ER | 34.33 (0.59) | 68.21 (1.27) | 40.24 (0.94) | 46.60 (1.36) | 57.46 (1.02) | 47.43 (1.79) |
| | BiC (Wu et al., 2019) | 35.96 (0.70) | 65.52 (0.96) | 42.06 (1.24) | 47.38 (0.54) | 55.72 (0.58) | 48.62 (0.42) |
| | LUCIR (Hou et al., 2019) | 38.40 (0.58) | 63.85 (0.42) | 42.14 (1.58) | 54.47 (1.06) | 40.94 (2.43) | 57.16 (0.86) |
| | PODNET (Douillard et al., 2020) | 26.19 (1.53) | 76.72 (2.14) | 35.68 (3.64) | 43.47 (2.81) | 60.73 (1.39) | 46.35 (1.14) |
| | DDE (Hu et al., 2021) | 36.16 (0.39) | 64.71 (0.81) | 39.74 (1.39) | 47.09 (0.58) | 55.92 (0.76) | 46.11 (1.03) |
| | DER++ (Buzzega et al., 2020) | 54.98 (0.23) | 46.33 (1.14) | 62.93 (1.03) | 61.70 (0.18) | 39.97 (0.43) | 60.23 (0.49) |
| 100 | CLSER (Arani et al., 2022) | 57.53 (0.48) | 42.85 (1.05) | 64.73 (2.66) | 57.38 (1.24) | 43.70 (0.41) | 61.92 (0.77) |
| | FOSTER (Wang et al., 2022a) | 30.62 (1.28) | / | / | 50.21 (0.15) | / | / |
| | MEMO (Zhou et al., 2023a) | 61.45 (0.49) | 37.87 (0.69) | 61.28 (1.10) | 62.77 (0.70) | 35.76 (1.44) | 58.81 (0.48) |
| | BEEF (Wang et al., 2023) | OOM | OOM | OOM | OOM | OOM | OOM |
| | BaCE w/o $Effect_{new}$&$Effect_{old}$ | 54.12 (0.29) | 46.06 (1.64) | 62.77 (0.52) | 61.80 (0.61) | 38.19 (1.58) | 59.28 (1.23) |
| | BaCE w/o $Effect_{new}$ | 62.85 (0.48) | 38.14 (1.21) | 67.11 (0.85) | 70.06 (0.39) | 30.51 (1.03) | 63.47 (1.42) |
| | BaCE w/o $Effect_{old}$ | 57.15 (0.62) | 43.55 (0.69) | 63.14 (1.60) | 64.28 (0.54) | 37.08 (0.89) | 61.48 (0.78) |
| | BaCE (Ours) | **65.88 (0.34)** | **34.05 (0.46)** | **69.53 (0.61)** | **74.81 (0.45)** | **24.78 (0.64)** | **67.14 (0.71)** |
| | ER | 70.68 (0.26) | 29.75 (0.24) | 67.16 (1.03) | 70.78 (0.42) | 30.28 (0.36) | 62.35 (0.66) |
| | BiC (Wu et al., 2019) | 72.86 (0.34) | 28.10 (0.41) | 68.07 (0.59) | 74.59 (0.13) | 24.84 (0.64) | 65.36 (1.52) |
| | LUCIR (Hou et al., 2019) | 73.27 (0.20) | 26.29 (0.48) | 69.93 (0.98) | 74.52 (0.41) | 21.68 (1.66) | 66.45 (1.34) |
| | PODNET (Douillard et al., 2020) | 31.10 (2.86) | 71.54 (5.27) | 41.54 (0.78) | 48.29 (2.01) | 55.42 (3.19) | 50.43 (2.84) |
| | DDE (Hu et al., 2021) | 74.23 (0.27) | 25.53 (0.44) | 70.19 (1.20) | 72.02 (0.58) | 28.43 (1.14) | 64.16 (0.58) |
| | DER++ (Buzzega et al., 2020) | 75.83 (0.35) | 23.72 (0.59) | 74.65 (0.64) | 75.17 (0.29) | 25.96 (0.61) | 66.26 (0.95) |
| 500 | CLSER (Arani et al., 2022) | 77.76 (0.82) | 21.50 (1.14) | 75.21 (1.42) | 78.54 (0.52) | 19.68 (0.35) | 68.35 (0.78) |
| | FOSTER (Wang et al., 2022a) | 70.21 (0.54) | / | / | 76.84 (0.29) | / | / |
| | MEMO (Zhou et al., 2023a) | 75.83 (0.42) | 22.56 (0.27) | 63.27 (0.59) | 78.96 (0.35) | 17.65 (0.46) | 61.72 (0.93) |
| | BEEF (Wang et al., 2023) | OOM | OOM | OOM | OOM | OOM | OOM |
| | BaCE w/o $Effect_{new}$&$Effect_{old}$ | 76.03(0.37) | 23.65 (1.41) | 74.59 (0.42) | 75.45 (0.31) | 24.91 (0.35) | 67.49 (0.35) |
| | BaCE w/o $Effect_{new}$ | 80.14 (0.28) | 17.34 (0.48) | 75.44 (0.65) | 82.13 (0.22) | 17.82 (0.51) | 69.14 (0.21) |
| | BaCE w/o $Effect_{old}$ | 78.54 (0.13) | 20.64 (0.54) | 74.12 (0.48) | 78.60 (0.25) | 20.87 (0.38) | 68.77 (0.79) |
| | BaCE (Ours) | **82.46 (0.25)** | **15.78 (0.63)** | **76.51 (0.66)** | **84.59 (0.20)** | **12.58 (0.33)** | **70.56 (0.48)** |
| ∞ | MTL | 91.67 (0.16) | / | / | 91.25 (0.13) | / | / |

**Full Results on ViT.** The full result with standard derivations is given in Table 7, 8 and 9. The evolution of average accuracy is shown in Fig. 12, 14 and 13. When we run the officially released code of BEEF (Wang et al., 2023) on CIFAR100 with two 24G RTX3090 GPUs, we encounter

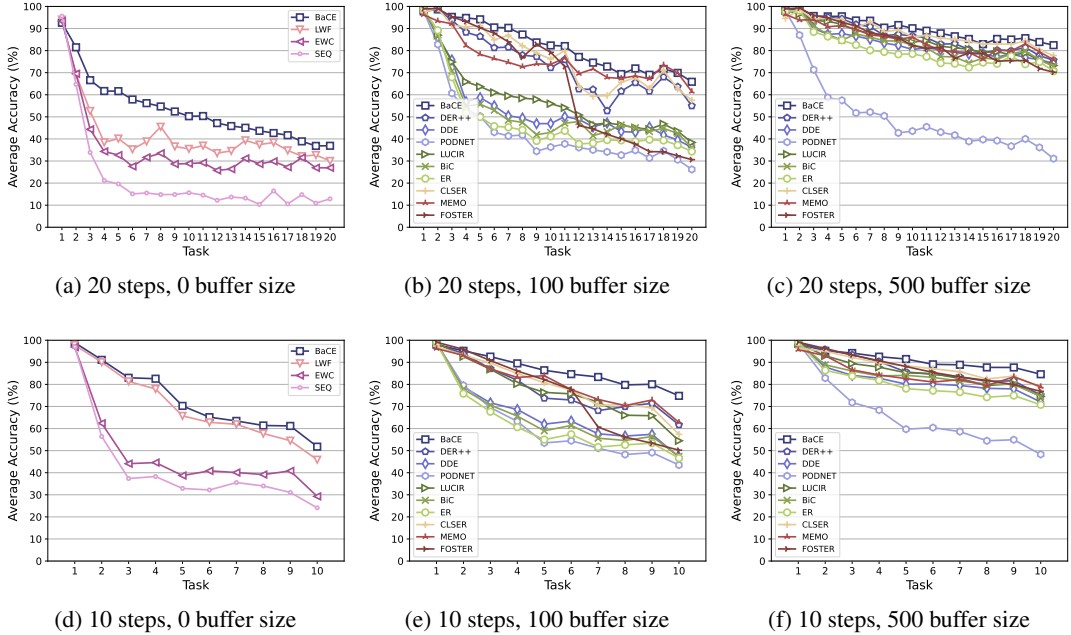

Figure 12: The evolution of average accuracy on split CIFAR-100 with ViT-B/16.

Table 8: The result on the 5-step split CIFAR-100. The backbone is pre-trained ViT-B/16. †: the baseline results are from Wang et al. (2022e).

| Buffer Size | Method | AverACC (↑) | Forgetting (↓) |
|---|---|---|---|
| 1000 | ER † | 67.87 (0.57) | 33.33 (1.28) |
| | LUCIR (Hou et al., 2019) | 79.69 (0.22) | 17.46 (0.60) |
| | PODNET (Douillard et al., 2020) | 64.19 (0.56) | 39.50 (0.52) |
| | DDE (Hu et al., 2021) | 77.39 (0.43) | 22.98 (0.75) |
| | DER++† (Buzzega et al., 2020) | 61.06 (0.87) | 39.87 (0.99) |
| | Gdumb † (Prabhu et al., 2020) | 67.14 (0.37) | / |
| | BiC † (Wu et al., 2019) | 66.11 (1.76) | 35.24 (1.64) |
| | Co2L † (Cha et al., 2021) | 72.15 (1.32) | 28.55 (1.56) |
| | L2P † (Wang et al., 2022e) | 84.21 (0.53) | 7.72 (0.77) |
| | BaCE (Ours) | 85.16 (0.39) | 6.86 (0.53) |
| 5000 | ER † | 82.53 (0.17) | 16.46 (0.25) |
| | LUCIR (Hou et al., 2019) | 84.58 (0.31) | 15.97 (0.35) |
| | PODNET (Douillard et al., 2020) | 67.74 (1.02) | 35.90 (1.59) |
| | DDE (Hu et al., 2021) | 84.05 (0.29) | 16.52 (1.22) |
| | DER++† (Buzzega et al., 2020) | 83.94 (0.34) | 14.55 (0.73) |
| | Gdumb † (Prabhu et al., 2020) | 81.67 (0.02) | / |
| | BiC † (Wu et al., 2019) | 81.42 (0.85)) | 17.31 (1.02) |
| | Co2L † (Cha et al., 2021) | 82.49 (0.89) | 17.48 (1.80) |
| | L2P † (Wang et al., 2022e) | 86.31 (0.59) | 5.83 (0.61) |
| | BaCE (Ours) | **88.43 (0.31)** | **4.33 (0.25)** |
| ∞ | MTL | 90.85 (0.12) | / |

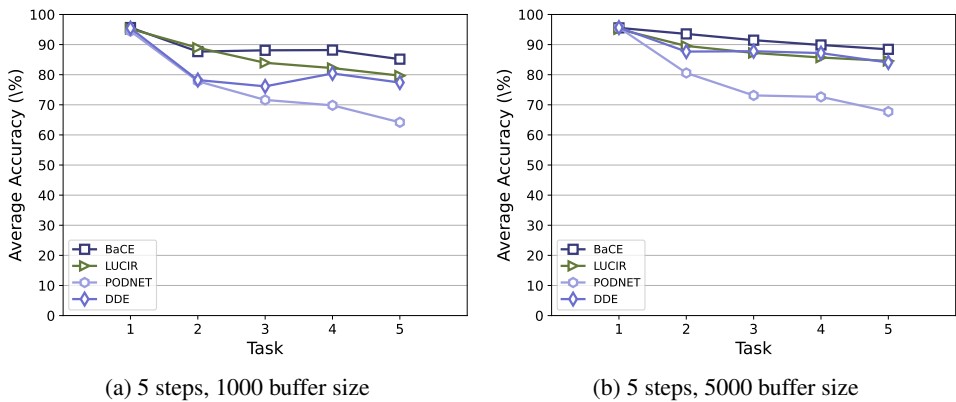

(a) 5 steps, 1000 buffer size  (b) 5 steps, 5000 buffer size

Figure 13: The evolution of average accuracy on 5-step split CIFAR100 with ViT-B/16.

Table 9: The comparison with competitive baselines on 5-datasets. All methods use pre-trained ViT-B/16 as the backbone. †: The baseline results are from Wang et al. (2022e).

| Buffer Size | Method | AverACC (↑) | Forgetting (↓) |
|---|---|---|---|
| **0** | FT-seq-frozen † | 39.49 (0.12) | 42.62 (0.20) |
| | FT-seq † | 20.12 (0.42) | 94.63 (0.68 |
| | EWC † (Kirkpatrick et al., 2017) | 50.93 (0.09) | **34.94 (0.07)** |
| | LwF † (Li & Hoiem, 2017) | 47.91 (0.33) | 38.01 (0.28 |
| | BaCE w/o $Effect_{new}$&$Effect_{old}$ | 21.37 (1.53) | 95.45 (0.61) |
| | BaCE w/o $Effect_{new}$ | 50.61 (0.81) | 47.49 (2.04) |
| | BaCE w/o $Effect_{old}$ | 30.69 (1.78) | 75.64 (3.25) |
| | BaCE (Ours) | **54.99 (0.69)** | 37.79 (1.47) |
| **250** | ER † | 80.32 (0.55) | 15.69 (0.89) |
| | Gdumb † (Prabhu et al., 2020) | 56.99 (0.06) | / |
| | BiC † (Wu et al., 2019) | 78.74 (1.41) | 21.15 (1.00) |
| | DER++ † (Buzzega et al., 2020) | 80.81 (0.07) | 14.38 (0.35) |
| | Co2L † (Cha et al., 2021) | 82.25 (1.17) | 17.52 (1.35) |
| | L2P † (Wang et al., 2022e) | 85.56 (0.95) | **4.22 (0.03)** |
| | CLSER (Arani et al., 2022) | 86.51 (0.36) | 10.58 (0.57) |
| | FOSTER (Wang et al., 2022a) | 74.21 (0.17) | / |
| | MEMO (Zhou et al., 2023a) | 86.53 (0.31) | 9.09 (0.36) |
| | BEEF (Wang et al., 2023) | 78.09 (0.22) | / |
| | BaCE w/o $Effect_{new}$&$Effect_{old}$ | 84.36 (0.62) | 9.58 (1.56) |
| | BaCE w/o $Effect_{new}$ | 86.10 (0.86) | 9.16 (0.97) |
| | BaCE w/o $Effect_{old}$ | 86.69 (0.40) | 10.58 (1.13) |
| | BaCE (Ours) | **87.50 (0.57)** | 8.41 (1.27) |
| **500** | ER † | 84.26 (0.84) | 12.85 (0.62) |
| | Gdumb † (Prabhu et al., 2020) | 70.76 (0.12) | / |
| | BiC † (Wu et al., 2019) | 85.53 (2.06) | 10.27 (1.32) |
| | DER++ † (Buzzega et al., 2020) | 84.88 (0.57) | 10.46 (1.02) |
| | Co2L † (Cha et al., 2021) | 86.05 (1.03) | 12.28 (1.44) |
| | L2P † (Wang et al., 2022e) | 88.95 (0.78) | **4.92 (0.71)** |
| | CLSER (Arani et al., 2022) | 89.43 (0.26) | 6.20 (0.75) |
| | FOSTER (Wang et al., 2022a) | 74.96 (0.30) | / |
| | MEMO (Zhou et al., 2023a) | 89.59 (0.15) | 5.37 (0.24) |
| | BEEF (Wang et al., 2023) | 79.13 (0.07) | / |
| | BaCE w/o $Effect_{new}$&$Effect_{old}$ | 86.20 (0.42) | 9.16 (1.32) |
| | BaCE w/o $Effect_{new}$ | 88.58 (0.63) | 5.64 (0.48) |
| | BaCE w/o $Effect_{old}$ | 88.79 (0.28) | 6.20 (0.71) |
| | BaCE (Ours) | **89.80 (0.27)** | 5.22 (0.76) |
| ∞ | MTL † | 93.93 (0.18) | / |

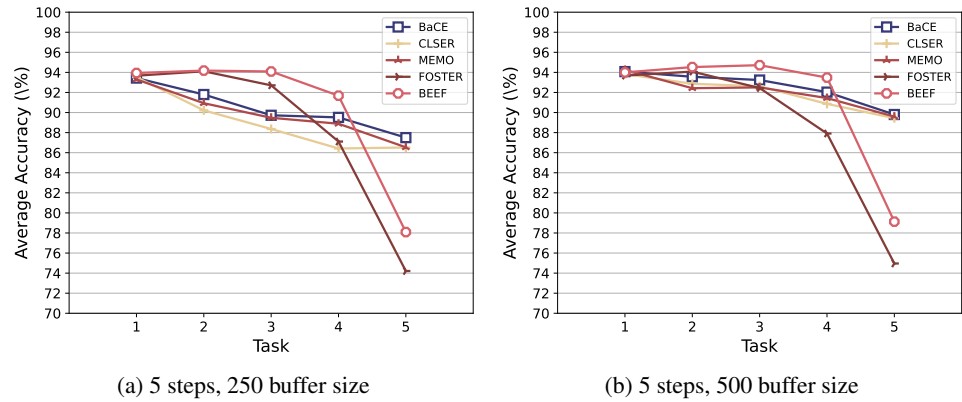

(a) 5 steps, 250 buffer size          (b) 5 steps, 500 buffer size

Figure 14: The evolution of average accuracy on 5-datasets with ViT-B/16.

the out-of-memory error. We also find that the architecture-based methods FOSTER and BEEF perform better than BaCE at the early stage of CIL. However, their performance degrades as more tasks are learned. We conjecture that the reason may be the hyper-parameters are not robust across datasets. Besides, we note that BaCE does not contain sensitive hyper-parameters and does not require additional GPU memory for each new task.

**Evaluation on DeiT-S/16 with Less Information Overlapping.** To mitigate information overlapping between pretraining data and the datasets for incremental learning, we follow the experimental setting in (Kim et al., 2023) to further evaluate the proposed method. We use DeiT-S/16 Touvron et al. (2021) as the backbone. We use the same model checkpoint as (Kim et al., 2023), which is trained using 611 classes of ImageNet after removing 389 classes which are similar or identical to the classes of CIFAR and Tiny-ImageNet. Following (Kim et al., 2023), we finetune adapters Houlsby et al. (2019) to leverage the strong performance of the pre-trained model while adapting to new knowledge.

For CIFAR10, 10 classes are split into 5 tasks with 2 classes for each task (C10-T5). For CIFAR100, 100 classes are split into 10 and 20 tasks (C100-T10 and C100-T200. For TinyImageNet, 200 classes are split into 5 and 10 tasks (T-5T and T-10T). Other experimental settings are consistent with (Kim et al., 2023). The experimental results are summarized in Table 10 and 11.

Table 10: The average accuracy after the final task. The buffer size for replay-based methods is 2000 for CIFAR100 and TinyImageNet and 200 for CIFAR10. The baseline results are from Table 2 of Kim et al. (2023).

| Method | C10-5T | C100-10T | C100-20T | T-5T | T-10T | Average |
|--------|--------|----------|----------|------|-------|---------|
| HAT | $79.36_{\pm5.16}$ | $68.99_{\pm0.21}$ | $61.83_{\pm0.62}$ | $65.85_{\pm0.60}$ | $62.05_{\pm0.55}$ | 67.62 |
| OWM | $41.69_{\pm6.34}$ | $21.39_{\pm3.18}$ | $16.98_{\pm4.44}$ | $24.55_{\pm2.48}$ | $17.52_{\pm3.45}$ | 24.43 |
| SLDA | $88.64_{\pm0.05}$ | $67.82_{\pm0.05}$ | $67.80_{\pm0.05}$ | $57.93_{\pm0.05}$ | $57.93_{\pm0.06}$ | 68.02 |
| PASS | $86.21_{\pm1.10}$ | $68.90_{\pm0.94}$ | $66.77_{\pm1.18}$ | $61.03_{\pm0.38}$ | $58.34_{\pm0.42}$ | 68.25 |
| L2P | $73.59_{\pm4.15}$ | $61.72_{\pm0.81}$ | $53.84_{\pm1.59}$ | $59.12_{\pm0.96}$ | $54.09_{\pm1.14}$ | 60.17 |
| iCaRL | $87.55_{\pm0.99}$ | $68.90_{\pm0.47}$ | $69.15_{\pm0.99}$ | $53.13_{\pm1.04}$ | $51.88_{\pm2.36}$ | 66.12 |
| A-GEM | $56.33_{\pm7.77}$ | $25.21_{\pm4.00}$ | $21.99_{\pm4.01}$ | $30.53_{\pm3.99}$ | $21.90_{\pm5.52}$ | 36.89 |
| EEIL | $82.34_{\pm3.13}$ | $68.08_{\pm0.51}$ | $63.79_{\pm0.66}$ | $53.34_{\pm0.54}$ | $50.38_{\pm0.97}$ | 63.59 |
| GD | $89.16_{\pm0.53}$ | $64.36_{\pm0.57}$ | $60.10_{\pm0.74}$ | $53.01_{\pm0.97}$ | $42.48_{\pm2.53}$ | 61.82 |
| DER++ | $84.63_{\pm2.91}$ | $69.73_{\pm0.99}$ | $70.03_{\pm1.46}$ | $55.84_{\pm2.21}$ | $54.20_{\pm3.28}$ | 66.89 |
| HAL | $84.38_{\pm2.70}$ | $67.17_{\pm1.50}$ | $67.37_{\pm1.45}$ | $52.80_{\pm2.37}$ | $55.25_{\pm3.60}$ | 65.39 |
| MORE | $89.16_{\pm0.96}$ | $70.23_{\pm2.27}$ | $70.53_{\pm1.09}$ | $64.97_{\pm1.28}$ | $63.06_{\pm1.26}$ | 71.59 |
| ROW | $90.97_{\pm0.19}$ | $\mathbf{74.72_{\pm0.48}}$ | $74.60_{\pm0.12}$ | $65.11_{\pm1.97}$ | $63.21_{\pm2.53}$ | 73.72 |
| BaCE | $\mathbf{91.54_{\pm0.31}}$ | $74.15_{\pm0.60}$ | $\mathbf{75.06_{\pm0.71}}$ | $\mathbf{66.23_{\pm1.28}}$ | $\mathbf{63.85_{\pm3.02}}$ | $\mathbf{74.02}$ |

Table 11: The average accuracy after the final task. The buffer size for replay-based methods is 1000 for CIFAR100 and TinyImageNet and 100 for CIFAR10. The baseline results are from Table 3 of Kim et al. (2023).

| Method | C10-5T | C100-10T | C100-20T | T-5T | T-10T | Average |
|--------|--------|----------|----------|------|-------|---------|
| iCaRL | $86.08_{\pm1.19}$ | $66.96_{\pm2.08}$ | $68.16_{\pm0.71}$ | $47.27_{\pm3.22}$ | $49.51_{\pm1.87}$ | 63.60 |
| A-GEM | $56.64_{\pm4.29}$ | $23.18_{\pm2.54}$ | $20.76_{\pm2.88}$ | $31.44_{\pm3.84}$ | $23.73_{\pm6.27}$ | 31.15 |
| EEIL | $77.44_{\pm3.04}$ | $62.95_{\pm0.68}$ | $57.86_{\pm0.74}$ | $48.36_{\pm1.38}$ | $44.59_{\pm1.72}$ | 58.24 |
| GD | $85.96_{\pm1.64}$ | $57.17_{\pm1.06}$ | $50.30_{\pm0.58}$ | $46.09_{\pm1.77}$ | $32.41_{\pm2.75}$ | 54.39 |
| DER++ | $80.09_{\pm3.00}$ | $64.89_{\pm2.48}$ | $65.84_{\pm1.46}$ | $50.74_{\pm2.41}$ | $49.24_{\pm5.01}$ | 62.16 |
| HAL | $79.16_{\pm4.56}$ | $62.65_{\pm0.83}$ | $63.96_{\pm1.49}$ | $48.17_{\pm2.94}$ | $47.11_{\pm6.00}$ | 60.21 |
| MORE | $88.13_{\pm1.16}$ | $71.69_{\pm0.11}$ | $71.29_{\pm0.55}$ | $64.17_{\pm0.77}$ | $61.90_{\pm0.90}$ | 71.44 |
| ROW | $89.70_{\pm1.54}$ | $73.63_{\pm0.12}$ | $71.86_{\pm0.07}$ | $65.42_{\pm0.55}$ | $62.87_{\pm0.53}$ | 72.70 |
| BaCE | $\mathbf{90.46_{\pm0.78}}$ | $\mathbf{73.76_{\pm0.64}}$ | $\mathbf{72.42_{\pm0.33}}$ | $\mathbf{65.59_{\pm1.02}}$ | $\mathbf{63.27_{\pm1.20}}$ | $\mathbf{73.10}$ |

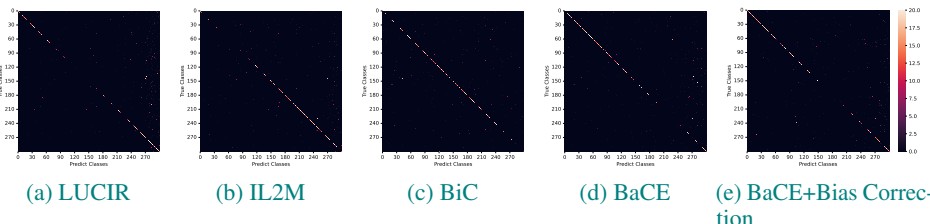

    (a) LUCIR      (b) IL2M      (c) BiC      (d) BaCE      (e) BaCE+Bias Correction

Figure 15: The confusion matrix of different methods on OmniBenchmark.

**Evaluation on Challenging Datasets.** We evaluate our model on four challenging datasets that have large domain gaps with ImageNet, namely ImageNet-R Hendrycks et al. (2021), ObjectNet Barbu et al. (2019), Omnibenchmark Zhang et al. (2022) and VTAB Zhai et al. (2019). ImageNet-R and ObjectNet contain challenging samples that ImageNet pre-trained models cannot handle Zhou et al. (2023b), while Omnibenchmark and VTAB contain diverse classes from multiple complex realms. We follow the experimental setting in (Zhou et al., 2023b) and use ViT-B/16-IN21K Dosovitskiy et al. (2020) as the backbone model, which is not supervised-finetuned on ImageNet1K. We use the datasets [4] sampled by Zhou et al. (2023b), where Omnibenchmark contains 300 classes, ImageNet-R and ObjectNet contain 200 classes, and VTAB contains 50 classes. We equally divide all classes into 10 tasks for each dataset.

We compare our method with several distillation-based methods: BiC Wu et al. (2019), IL2M Belouadah & Popescu (2019), LUCIR Hou et al. (2019), DER++ Buzzega et al. (2020), CLSER Arani et al. (2022). All baselines and BaCE store 5 samples for each class on ObjectNet, Omnibenchmark and VTAB and store 20 samples for each class on ImageNet-R. All baselines and BaCE load the same PTM (i.e., ViT-B/16-IN21K) for training. We set $\alpha = 2$ for BaCE on the four datasets. The results are summarize in Table 12.

Table 12: The average accuracy (Acc.), the difference of feature embedding distance between new classes and old classes (Feat.Embed.Dist.) and the probing accuracy (Prob.Acc.) after learning the final task on four challenging datasets.

| | OmniBenchmark | | | ImageNet-R | | | ObjectNet | | | VTAB | | |
|---|---|---|---|---|---|---|---|---|---|---|---|---|
| | Acc. | Prob.Acc | Feat.Embed.Dist | Acc. | Prob.Acc | Feat.Embed.Dist | Acc. | Prob.Acc | Feat.Embed.Dist | Acc. | Prob.Acc | Feat.Embed.Dist |
| ER | 68.50 | 74.51 | 0.1376 | 65.88 | 72.28 | 0.0731 | 46.14 | 52.14 | 0.1485 | 78.53 | 90.37 | 0.1954 |
| BiC | 71.84 | 74.36 | 0.0709 | 70.14 | 73.13 | 0.0782 | 48.65 | 52.63 | 0.1377 | 81.65 | 90.15 | 0.1732 |
| LUCIR | 67.18 | 73.85 | 0.1621 | 67.43 | 71.06 | 0.0956 | 47.39 | 52.77 | 0.1365 | 80.49 | 89.79 | 0.1814 |
| IL2M | 69.52 | 74.94 | 0.1230 | 69.39 | 74.70 | 0.0505 | 48.42 | 52.55 | 0.1106 | 80.37 | 90.55 | 0.1623 |
| DER++ | 72.11 | 74.68 | 0.1156 | 72.18 | 75.58 | 0.0476 | 49.18 | 52.61 | 0.0974 | 79.09 | 90.62 | 0.1836 |
| CLSER | 72.24 | 74.13 | 0.1235 | 71.93 | 76.02 | 0.0670 | 51.04 | 52.38 | 0.1182 | 79.56 | 90.51 | 0.1878 |
| BaCE | 73.30 | 75.17 | 0.0392 | 74.48 | 77.55 | 0.0345 | 52.29 | 53.30 | 0.0825 | 84.86 | 90.84 | 0.0652 |
| BaCE + Bias Correction | **73.93** | **75.17** | **0.0392** | **75.20** | **77.55** | **0.0345** | **53.32** | **53.30** | **0.0825** | **85.53** | **90.84** | **0.0652** |

[4]https://github.com/zhoudw-zdw/RevisitingCIL

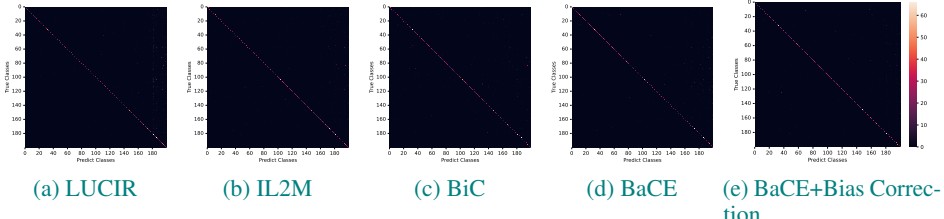

(a) LUCIR     (b) IL2M     (c) BiC     (d) BaCE     (e) BaCE+Bias Correction

Figure 16: The confusion matrix of different methods on ImageNetR.

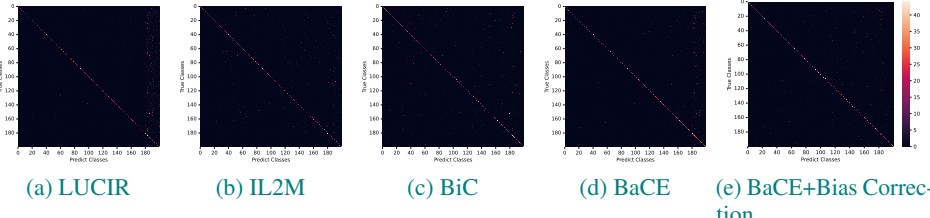

(a) LUCIR     (b) IL2M     (c) BiC     (d) BaCE     (e) BaCE+Bias Correction

Figure 17: The confusion matrix of different methods on ObjectNet.

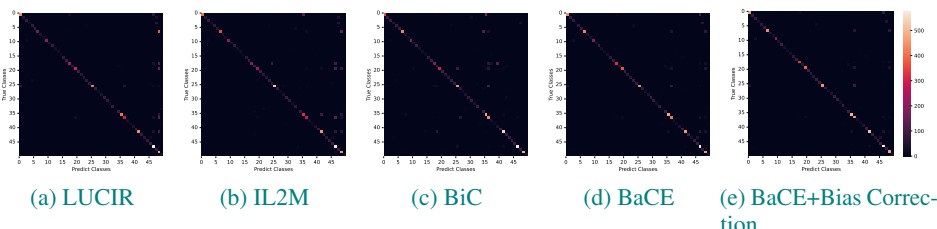

(a) LUCIR     (b) IL2M     (c) BiC     (d) BaCE     (e) BaCE+Bias Correction

Figure 18: The confusion matrix of different methods on VTAB.

**The difference between causal imbalance problem and class imbalance problem.** The class imbalance problem is a long-standing problem in CIL. It refers to the phenomenon that models tend to predict new classes. The class imbalance problem is caused by the imbalance in the number of samples between new and old tasks. However, the causal imbalance problem (i.e., the confrontational phenomenon) is different from the class imbalance problem. As mentioned in Appendix A.4, the causal imbalance problem focuses on the learning process of both new and old classes. In contrast, the class imbalanced problem only focuses on the scale of logits.

What is the benefit of addressing the causal imbalance problem instead of the class imbalance problem? As shown in Table 12, addressing the causal imbalance problem reduces the feature embedding distance between new and old classes significantly. More importantly, it improves the probing accuracy over existing methods designed for the class imbalance problem. It indicates that only balancing the scale of logits (e.g., BiC and IL2M) may not necessarily enhance the feature learning process. In contrast, BaCE pursues causal-balanced learning for each class, thereby achieving higher probing performance in IL.

To have a closer look at the class imbalance problem, we provide the confusion matrices of BaCE, BiC, LUCIR, IL2M on the four challenging datasets in Fig. 15, Fig. 16, Fig. 17, and Fig. 18. BiC shows the most balanced predictions, followed by BaCE and IL2M, and finally, LUCIR. However, BiC and IL2M make more errors in predicting old classes and perform lower than BaCE. We speculate that the reason behind this is that the old rehearsal data does not reflect the true distribution of the old data, and pursuing a balanced prediction between the rehearsal data and new data may hurt the representation ability during IL.

Table 13: The result on the 10-step split CIFAR-100. All methods use randomly initialized ResNet-18 as the backbone.

| Buffer Size | Method | 10 steps | |
| --- | --- | --- | --- |
| | | AverACC (↑) | FGT(↓) |
| 0 | SEQ | 9.43 | 89.82 |
| | LwF (Li & Hoiem, 2017) | 16.22 | 54.89 |
| | BaCE (Ours) | **19.07** | **47.85** |
| 500 | ER | 22.10 | 73.64 |
| | BiC (Wu et al., 2019) | 36.02 | 51.85 |
| | iCaRL (Rebuffi et al., 2017) | 46.52 | 22.06 |
| | LUCIR (Hou et al., 2019) | 40.59 | 34.55 |
| | DER (Buzzega et al., 2020) | 36.60 | 54.99 |
| | DER++ (Buzzega et al., 2020) | 38.25 | 50.54 |
| | BaCE (Ours) | **42.18** | **43.11** |
| 2000 | ER | 38.58 | 53.58 |
| | BiC (Wu et al., 2019) | 46.39 | 40.49 |
| | iCaRL (Rebuffi et al., 2017) | 49.82 | 18.07 |
| | LUCIR (Hou et al., 2019) | 41.73 | 25.41 |
| | DER (Buzzega et al., 2020) | 51.89 | 34.54 |
| | DER++ (Buzzega et al., 2020) | 53.63 | 33.66 |
| | BaCE (Ours) | **57.73** | **23.53** |
| ∞ | MTL | 70.44 | / |

The confusion matrices also show that BiC has a more balanced prediction than BaCE. Therefore, we rectify the prediction bias towards new classes in BaCE using the re-balancing method in BiC. Specifically, we follow BiC to learn two additional parameters to correct the bias on the balanced datasets and select the best parameters according to the validation set. We only re-balance the prediction after learning the last task and the final model is denoted as BaCE+Bias Correction. The learned bias parameters $(\alpha, \beta)$ on OmniBenchmark, ImageNetR, ObjectNet, and VTAB are $(0.94423, -0.01378)$, $(0.93485, -0.06928)$, $(0.84370, -0.09861)$, and $(0.92134, -0.05441)$ respectively. As shown in Table 12, correcting the bias slightly improves performance. Since the encoder and the classifier are unchanged, the Prob.Acc. and the Feat.Embed.Dist remains unchanged. The confusion matrices show that BaCE+Bias Correction achieves a more balanced prediction.

In summary, the causal imbalance problem and the class imbalance problem are different since the former describes the learning process while the latter describes the scale of prediction logits. More importantly, these two problems are not conflicting, and the technique for the class imbalance problem can be combined with BaCE to improve the performance further. Analyzing the causalities in CIL helps us to clarify the reason behind forgetting. The causal perspective helps us design algorithms that better maintain the model representation ability in IL. In contrast, the methods for the class imbalance problem superficially balance the scale of logits, and thus, they are difficult to guide the model to learn better features.

**Experiments on ResNet-18.** Although our motivation is based on PTMs, BaCE is not limited to CIL with PTMs. We apply BaCE to ResNet-18 with the same hyper-parameters as in ViT. The result in Table 13 shows that BaCE also outperforms competitive CIL methods consistently. We notice that the performance gain of BaCE becomes smaller when using ResNet-18 as the backbone. The reason may be that BaCE closes the gap to probing performance by addressing the confrontation phenomenon, but the probing performance of ResNet-18 is much lower than that of ViT. Moreover, it indicates that mitigating the confrontation phenomenon is beneficial not only to pre-trained models but also to generic models.

**KNN Examples in *Effect*$_{new}$.** We provide more KNN examples on CIFAR100 and 5-datasets in Fig. 19. The ground-truth label is on the top of each image. The green and red labels of neighbors represent whether or not they are the same as the input sample. Fig. 19 shows that the teacher

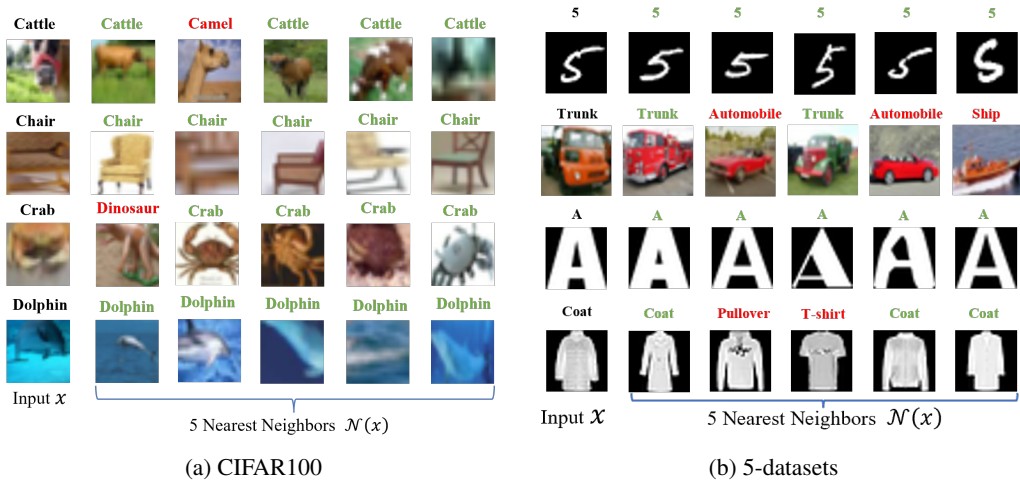

(a) CIFAR100          (b) 5-datasets

Figure 19: The KNNs in $Effect_{new}$ on CIFAR100 and 5-datasets

model can recognize new classes with prior knowledge to some extent, which is an intuitive way to understand how $Effect_{new}$ consider the causal effects from both new and old data.

**The effect of KNNs in $Effect_{new}$**     To further explore the role of KNNs in $Effect_{new}$, we consider the following variants: (1) only the neighbours with the same ground-truth labels as the input sample are selected (denoted as *w/ same labels*); (2) only the neighbours with the different ground-truth labels from the input sample are selected (denoted as *w/ different labels*); (3) the weights of all neighbours are set as the same (denoted as *w/ same weight*); (4) the neighbours are randomly sampled from all training data (denoted as *w/ random neighbours*); (5) the neighbours with the largest distance are selected (denoted as *w/ largest distance*);

Table 14 shows that all ablated variants perform worse than BaCE. Besides, as shown in Fig. 19, some neighbours have different labels as the input sample. It indicates that the teacher model may provide incorrect information about new classes. However, the performance decreases when we filter out the neighbours with different labels (i.e., *w/ same labels*). The results show that the KNNs act as a constraint from the teacher model and thus build causal paths from both new and old data when the student models adapt to new classes. For example, in Fig. 19a, the teacher model provides the prior that cattle and camel are animals walking on land.

Table 14: The effect of KNNs in $Effect_{new}$. The average accuracy after the final task on the four challenging datasets is reported.

|  | OmniBenchmark | ImageNet-R | ObjectNet | VTAB |
|---|---|---|---|---|
| BaCE | **73.30** | **74.48** | **52.29** | **84.86** |
| w/ same labels | 72.73 | 73.59 | 50.68 | 83.26 |
| w/ different labels | 72.81 | 72.94 | 50.15 | 82.81 |
| w/ same weight | 73.02 | 73.97 | 51.66 | 84.22 |
| w/ random neighbours | 72.88 | 73.41 | 50.66 | 83.01 |
| w/ largest distance | 72.42 | 72.94 | 50.56 | 82.83 |
| w/o $Effect_{new}$ | 72.82 | 73.46 | 50.78 | 83.17 |

**Runtime and Hyper-parameter Analysis.**     We summarize the average accuracy of BaCE when different $K$ is selected in Table 15. The experiment is conducted on a 20-step split CIFAR100 with 500 replay instances using ViT as the backbone. Table 15 shows that BaCE achieves the best performance when $K = 5$ and the performance is degraded when $K = 10$. It indicates that selecting more neighbors is beneficial when $K <= 5$. However, estimating $Effect_{new}$ with a large $K$ may bring noise to the learning process.

Table 15: The hyper-parameter analysis of $K$.

| Method | REPLAY | BaCE (K=0) | BaCE (K=1) | BaCE (K=2) | BaCE (K=5) | BaCE (K=10) |
|---|---|---|---|---|---|---|
| **AverACC** | 70.68 | 80.14 | 81.38 | 82.11 | **82.46** | 81.04 |
| **Run Time** | $\times 1.0$ | $\times 1.2$ | $\times 2.5$ | $\times 2.7$ | $\times 3.2$ | $\times 5.2$ |

We compare the runtime of BaCE and REPLAY. The result indicates that BaCE takes a longer training time than REPLAY. The reason is that the neighbor relationship and the forward passes of neighbors are computed.

## D.2 CONTINUAL TEXT CLASSIFICATION (CONTINUAL TC)

**Training Settings.** We opt for bert-base-cased (Devlin et al., 2019; Wolf et al., 2019) as the backbone model for all approaches since its popularity in NLP and the pre-trained weights are downloaded from Huggingface (Wolf et al., 2019). We train each model for 3 epochs and tune the batch size in 8, 16, the learning rate of the backbone model in 1e-4, 3e-5, 1e-5. The learning rate of the final linear classifier is set as 1e-3, and the maximum sequence length is set as 128.

**Hyper-parameters** In $Effect_{new}$, we fix the number of neighbors $K = 5$ and the weight $W_0 = 0.95$. In $Effect_{old}$, we fix $\alpha = 2$ when no buffer is available and $\alpha = 1$ when a buffer is available.

**Datasets.** We use three publicly available topic classification datasets, including AGNews, DBPedia, and Yahoo (Zhang et al., 2015), which are collected from various domains. Different from de Masson D'Autume et al. (2019), we do not use Yelp and Amazon since their labels are product ratings from 1-5, which simplifies CIL to Task-Incremental Learning. Besides, we divide DBPedia and Yahoo into two sub-datasets with disjoint label sets, respectively. In total, we obtained five datasets, including TC AGNews, DBPedia1, DBPedia2, Yahoo1, and Yahoo2. Since we focus on CIL, we remove the overlapping labels to ensure the label set between datasets is disjoint. Then, the categories in each dataset are as follows: AGNews (4 classes including: *World*, *Sports*, *Business*, *Sci_Tech*); DBPedia1 (6 classes including: *Artist*, *Athlete*, *OfficeHolder*, *MeanOfTransportation*,*Building*, *NaturalPlace*); DBPedia2 (6 classes including: *Village*, *Animal*, *Plant*, *Album*, *Film*, *WrittenWork*); Yahoo1 (3 classes including: *Society_Culture*,*Health*,*Education_Reference*); Yahoo2 (3 classes including: *Computers_Internet*, *Family_Relationships*, *Politics_Government*). Following de Masson D'Autume et al. (2019), we randomly sample an equal number of training samples for each category. Concretely, each category contains 28,000 training samples and 2,000 test samples. We define the task sequence as follows: AGNews→DBPedia1→DBPedia2→Yahoo1→Yahoo2.

**Baselines.** For continual TC, we compare the following strong baselines: LwF (Li & Hoiem, 2017), EWC (Kirkpatrick et al., 2017), Sparse-ER (de Masson D'Autume et al., 2019), ER (Chaudhry et al., 2019), MBPA (Sprechmann et al., 2018), MBPA++ (de Masson D'Autume et al., 2019), IDBR (Huang et al., 2021).

- Sparse-ER (de Masson D'Autume et al., 2019): Sparse-ER replays stored samples at sparse intervals (e.g. 100 steps).

- MBPA (Sprechmann et al., 2018): MBPA uses stored examples for local adaptation without sparse experience replay. We set the number of neighbors K = 32 and the number of local adaptation steps L = 30.

- MBPA++ (de Masson D'Autume et al., 2019): MBPA++ adopts Sparse-ER and MBPA to learn new knowledge and reuse previously acquired knowledge. The original implementation of MBPA++ utilizes Sparse-ER, which stores 1% of the total training data seen so far. However, we find that the performance of Sparse-ER degrades significantly when the buffer size is fixed and limited. Therefore, we adopt ER for MBPA++ instead of Sparse-ER. We set the number of neighbors K = 32 and the number of local adaptation steps L = 30.

- IDBR (Huang et al., 2021): IDBR is an information disentanglement-based regularization method for continual text classification. In its original implementation, the buffer stores a fixed ratio of all training samples seen so far. For a fair comparison, we fix the buffer size as 100 and 500, respectively, in our experiments. We use the same default hyper-parameters as in the officially released code in `https://github.com/GT-SALT/IDBR`.

Table 16: The results of continual TC. All methods use pre-trained bert-based-cased as the backbone.

| Buffer Size | Method | AverACC (↑) | FGT (↓) | FWT (↑) |
|---|---|---|---|---|
| 0 | SEQ | 21.19 | 87.25 | 27.97 |
| | EWC (Kirkpatrick et al., 2017) | 24.84 | 82.50 | 28.83 |
| | LwF (Li & Hoiem, 2017) | 36.17 | 68.39 | 38.96 |
| | BaCE (Ours) | **58.75** | **30.66** | **44.05** |
| 100 | Sparse-ER (de Masson D'Autume et al., 2019) | 63.71 | 34.12 | 51.90 |
| | ER | 69.58 | 26.54 | 53.56 |
| | MBPA (Sprechmann et al., 2018) | 61.06 | 29.78 | 29.04 |
| | MBPA++ (de Masson D'Autume et al., 2019) | 74.11 | 17.64 | 53.28 |
| | IDBR (Huang et al., 2021) | 80.82 | 10.55 | 52.77 |
| | BaCE (Ours) | **81.56** | **9.70** | **54.04** |
| 500 | Sparse-ER (de Masson D'Autume et al., 2019) | 64.16 | 33.40 | 52.13 |
| | ER | 77.44 | 16.62 | 52.60 |
| | MBPA (Sprechmann et al., 2018) | 61.28 | 30.66 | 27.37 |
| | MBPA++ (de Masson D'Autume et al., 2019) | 78.72 | 12.70 | 53.73 |
| | IDBR (Huang et al., 2021) | 83.12 | 10.43 | 53.44 |
| | BaCE (Ours) | **83.53** | **8.28** | **54.21** |
| ∞ | MTL | 86.59 | / | / |

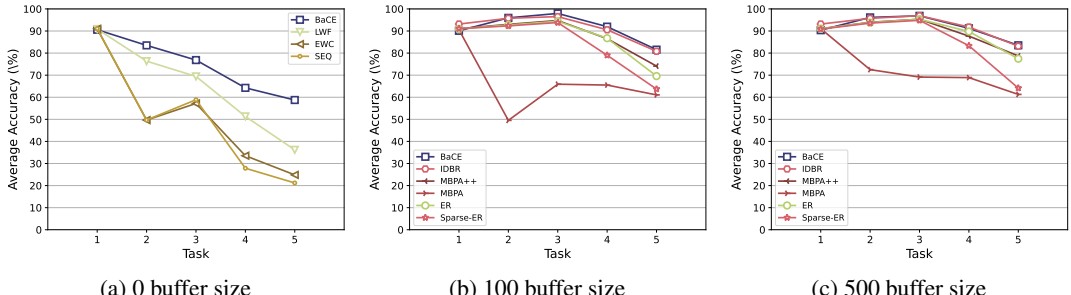

| (a) 0 buffer size | (b) 100 buffer size | (c) 500 buffer size |
|---|---|---|

Figure 20: The evolution of average accuracy of continual TC.

**Results.** The result of continual TC is summarized in Table. 16 and Fig. 20. The result indicates that BaCE outperforms not only CIL methods for language PTMs, including MBPA++ and Sparse-ER, but also task-specific CIL methods, IDBR. **Results.**

Table 17: The KNNs in $Effect_{new}$ on text classification datasets.

| | Label | Text |
|---|---|---|
| Input Sample | Artist | Hong 10 Hong 10 ( Birth name : Kim Hong - Yeol Korean : born on 27 December 1984 in South Korea ) is a male South Korean B - boy ( also known as Breakdancer ). His accomplishments include two Red Bull BC One individual titles ( 2006 ) & ( 2013 ) and a Battle of the Year crew title ( 2002 ). |
| KNN-1 | Artist | Dave Parsons Dave Parsons ( born David Richard Parsons 2 July 1965 Hillingdon London UK ) is a British bass guitarist. |
| KNN-2 | Artist | Porter Wagoner Porter Wayne Wagoner ( August 12 1927 – October 28 2007 ) was a popular American country music singer known for his flashy Nudie and Manuel suits and blond pompadour. In 1967 he introduced then - obscure singer Dolly Parton on his long - running television show and they were a well - known vocal duo throughout the late 1960s and early 1970s. Known as Mr. Grand Ole Opry Wagoner charted 81 singles from 1954 – 1983. He was elected to the Country Music Hall of Fame in 2002. |
| KNN-3 | OfficeHolder | Daylin Leach Daylin Leach ( born June 23 1961 ) is a Democratic member of the Pennsylvania State Senate who has represented the 17th senatorial district since 2009. He was previously a member of the Pennsylvania House of Representatives representing the 149th district from 2003 to 2009. On April 2 2013 he announced his candidacy for the U. S. |
| KNN-4 | Artist | Bebo Norman Jeffrey Stephen Bebo Norman ( born May 29 1973 ) is a contemporary Christian musician from Columbus Georgia USA. His most successful album to date is Myself When I Am Real which included hit songs Great Light of the World and Falling Down. Other popular songs by Norman include Disappear Nothing Without You I Will Lift My Eyes and Borrow Mine. He initially gained popularity when touring with another Christian band Caedmon's Call. Norman's fans call themselves Simpletons. |
| KNN-5 | OfficeHolder | James Arciero James Arciero is an American state legislator serving in the Massachusetts House of Representatives. |
| | Label | Text |
| Input Sample | Plant | Ephedra pedunculata Ephedra pedunculata common name Clap - weed vine Mormon tea or Comida de Vívora is a plant species native to southern Texas and to Mexico as far south as Zacatecas. It grows in sandy or rocky slopes and outcrops. Most species of Ephedra ( called Mormon tea ) are shrubs but Ephedra pedunculata is a trailing or clambering woody vine up to 7 m ( 23 feet ) long. Bark is gray becoming cracked with age. Leaves are opposite up to 3 mm ( 0. 12 inches ) long. |
| KNN-1 | Plant | Tetragonia decumbens Tetragonia decumbens commonly known as sea spinach is a coastal shrub native to southern Africa. |
| KNN-2 | Animal | Long - whiskered Owlet The Long - whiskered Owlet ( Xenoglaux loweryi ) is a tiny owl that is endemic to a small area in the Andean mountains in Amazonas and San Martín in northern Peru. It is restricted to cloud forests with dense undergrowth and epiphytes at about 1890 – 2200 meters ( 6200 – 7220 ft ) above sea level. The Long - whiskered Owlet is mainly brown with a whitish belly and eyebrow. The large eyes are orange - brown. |
| KNN-3 | Plant | Acacia salicina Acacia salicina is a thornless species of Acacia tree native to Australia. Common names include Cooba Native Willow Willow Wattle Broughton WillowSally Wattle and Black Wattle. It is a large shrub or small evergreen tree growing 3 to 20 m tall. It has a life span of about 10 – 15 years. In the Northern Hemisphere Acacia salicina flowers primarily from October to January and the seed pods are often visible from April to July. |
| KNN-4 | Village | Chipring Chipring is a village and Village Development Committee in Khotang District in the Sagarmatha Zone of eastern Nepal. At the time of the 1991 Nepal census it had a population of 1331 persons living in 263 individual households. |
| KNN-5 | Plant | Jacqueshuberia pustulata Jacqueshuberia pustulata is a plant species endemic to Venezuela. It is known only from a single location along a blackwater stream in the State of Amazonas at an elevations of about 115 m. Jacqueshuberia pustulata is a tree up to 5 m tall. Stipules are compound with up to 20 pairs of leaflet - like lobes each up to 9 mm long. |

**KNN Examples in $Effect_{new}$.** We provide more KNN examples on text classification datasets in Table 17.

### D.3 CONTINUAL NAMED ENTITY RECOGNITION (CONTINUAL NER)

**Training Settings.** We use bert-base-cased (Devlin et al., 2019; Wolf et al., 2019) as the backbone model since its popularity. We train each model for 5 epochs and tune the batch size in 8, 16, the learning rate of the backbone model in 1e-4, 3e-5, 1e-5. The learning rate of the final linear classifier is set as 1e-3, and the maximum sequence length is set as 256.

**Hyper-parameters** In $Effect_{new}$, we fix the number of neighbors $K = 5$ and the weight $W_0 = 0.95$. In $Effect_{old}$, we fix $\alpha = 2$ when no buffer is available and $\alpha = 1$ when a buffer is available.

**Datasets.** We select five commonly-used NER datasets for continual NER, including CoNLL2003 (Sang & De Meulder, 2003), I2B2 (Murphy et al., 2010), MIT Restaurant (Liu et al., 2013a), MIT Movie (Liu et al., 2013b), OntoNotes5 (Hovy et al., 2006). Compared with continual TC, continual NER is a more practical and challenging scenario because the number of classes between tasks and samples between classes is imbalanced. For example, the number of entities in OntoNotes5 and

Table 18: The statistics of the NER datasets

| Datasets | # Entity Types | # Train Sentences | # Test Sentences | Entities |
|---|---|---|---|---|
| CoNLL2003 | 4 | 11.1 | 2.8 | LOCATION, MISC, ORGANISATION, PERSON |
| MIT Restaurant | 7 | 7.3 | 1.5 | Amenity, Cuisine, Dish, Hours, Price, Rating, Restaurant_Name |
| I2B2 | 10 | 3.0 | 2.8 | AGE, HOSPITAL, IDNUM, MEDICALRECORD, PHONE, PROFESSION, STATE, STREET, USERNAME, ZIP |
| MIT Movie | 12 | 7.8 | 1.9 | Actor, Award, Character_Name, Director, Genre, Opinion, Origin, Plot, Quote, Relationship, Soundtrack, Year |
| OntoNotes5 | 14 | 21.0 | 2.9 | CARDINAL, DATE, EVENT, FAC, LANGUAGE, LAW, MONEY, NORP, ORDINAL, PERCENT, PRODUCT, QUANTITY, TIME, WORK_OF_ART |

CoNLL2003 is 14 and 4. In the training set of OntoNotes5, there are 10922 *DATE* entities but only 282 *LAW* entities. We define the task sequence as follows: OntoNotes5→MIT Movie→I2B2→MIT Restaurant→CoNLL2003. The dataset statistics are summarized in Table 18.

**Baselines.** For continual NER, we select the following competitive methods: LwF (Li & Hoiem, 2017), EWC (Kirkpatrick et al., 2017), Sparse-ER (de Masson D'Autume et al., 2019), ER (Chaudhry et al., 2019), MBPA (Sprechmann et al., 2018), MBPA++ (de Masson D'Autume et al., 2019), ExtendNER (Monaikul et al., 2021), CFNER (Zheng et al., 2022). For LwF, EWC, Sparse-ER, ER, MBPA, and MBPA++, we use the same setting as in continual IC and continual TC. For task-specific methods, the introductions are as follows:

- ExtendNER: ExtendNER is a distillation-based method. Specifically, ExtendNER computes the cross-entropy loss of entity tokens and the Kullback-Leibler Divergence loss of Other-class tokens. During training, the sum of the cross-entropy loss and KL divergence loss is minimized jointly. The temperature of the student model is set as 2.

- CFNER: CFNER is built upon ExtendNER. Unlike ExtendNER, CFNER further computes the causal effects from both entity and Other-class tokens for mitigating the catastrophic forgetting caused by Other-class tokens. We use the same default hyper-parameters as in the official released code in `https://github.com/zzz47zzz/CFNER`.

**Results.** The result of continual NER is summarized in Table. 19 and Fig. 21. The result indicates that BaCE consistently outperforms generic methods, including LwF, EWC, Sparse-ER, ER, MBPA, and MBPA++, as well as task-specific methods, including ExtendNER and CFNER.

Table 19: The results of continual NER. All methods use pre-trained bert-based-cased as the backbone.

| Buffer Size | Method | AverACC (↑) | FGT (↓) | FWT (↑) |
|---|---|---|---|---|
| 0 | SEQ | 19.58 | 60.40 | 26.20 |
| | EWC Kirkpatrick et al. (2017) | 22.65 | 55.96 | 27.32 |
| | LwF Li & Hoiem (2017) | 24.82 | 52.65 | 30.80 |
| | ExtendNER Monaikul et al. (2021) | 50.56 | 21.78 | 31.44 |
| | CFNER Zheng et al. (2022) | 54.10 | 17.07 | 34.32 |
| | BaCE (Ours) | **56.54** | **15.73** | **41.73** |
| 100 | Sparse-ER de Masson D'Autume et al. (2019) | 29.83 | 48.14 | 28.69 |
| | ER | 46.49 | 26.70 | 33.26 |
| | MBPA Sprechmann et al. (2018) | 38.56 | 30.64 | 26.32 |
| | MBPA++ de Masson D'Autume et al. (2019) | 45.37 | 23.56 | 32.10 |
| | ExtendNER Monaikul et al. (2021) | 51.69 | 22.94 | 37.8 |
| | CFNER Zheng et al. (2022) | 56.16 | 14.61 | 34.87 |
| | BaCE (Ours) | **58.33** | **10.97** | **42.37** |
| 500 | Sparse-ER de Masson D'Autume et al. (2019) | 30.81 | 46.34 | 46.34 |
| | ER | 57.17 | 13.54 | 36.98 |
| | MBPA Sprechmann et al. (2018) | 38.98 | 28.00 | 26.11 |
| | MBPA++ de Masson D'Autume et al. (2019) | 53.96 | 14.50 | 36.96 |
| | ExtendNER Monaikul et al. (2021) | 58.06 | 13.69 | 42.60 |
| | CFNER Zheng et al. (2022) | 60.08 | 12.76 | 42.24 |
| | BaCE (Ours) | **62.18** | **7.78** | **43.98** |
| ∞ | MTL | 71.62 | / | / |

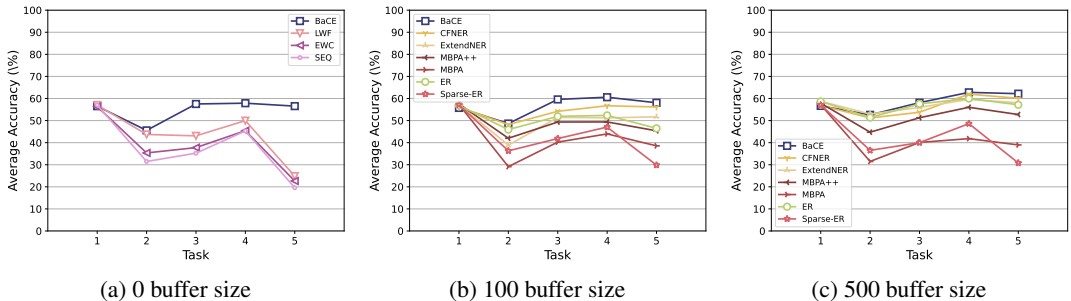

(a) 0 buffer size       (b) 100 buffer size       (c) 500 buffer size

Figure 21: The evolution of average accuracy of continual NER.

Table 20: The KNNs in $Effect_{new}$ on named entity recognition datasets.

| | Label | Word | Sentence |
|---|---|---|---|
| Input Sample | B-Location | Asia | Both sides also exchanged views on relevant problems such as the security situation and economic development in Asia, the unification of Europe, etc. |
| KNN-1 | B-Location | Asia | Peres said : "China and Israel are at the two ends of Asia , separated by 'ten thousand crags and torrents .' |
| KNN-2 | B-Location | Asia | He said in addition : "We who live in western Asia are gazing at the east with a look of hope and reverence . |
| KNN-3 | I-Location | Asia | And if you are in South Asia , perhaps you heard the Radio Canada International test transmissions to your part of the world back on the 8th , 9th , and 10th of November . |
| KNN-4 | B-Location | Asia | Eh you know we've got a very strong military force and deterrent force at work in Asia and particularly on the Korean peninsula . |
| KNN-5 | B-Location | Asia | Currently , TGS has its sights set mainly on doing research into diseases which are especially prevalent in Asia , such as liver and stomach cancers . |

| | Label | Word | Sentence |
|---|---|---|---|
| Input Sample | B-Dish | pizza | would you be able to find a place that has takeout pizza in my area |
| KNN-1 | B-Cuisine | pizza | are there any good pizza places around here |
| KNN-2 | B-Dish | pizza | are there any kid friendly pizza parlors around here |
| KNN-3 | B-Dish | pizza | are there any pizza places that are still open after midnight less than 10 minutes from here |
| KNN-4 | I-Dish | pizza | are there any pizzerias on long island that 1 slices of pizza |
| KNN-5 | B-Dish | pizza | can you find a pizza place |

**KNN Examples in *Effect*$_{new}$.** We provide more KNN examples on named entity recognition datasets in Table 20.

