# OpenReview forum: "Classifiers are Forgetful! Balancing the Mutual Causal Effects in Class-Incremental Learning"
_ICLR.cc/2024/Conference — Submitted to ICLR 2024_

### Official Review · Reviewer_6Ezu · 2023-10-27

**Soundness:** 2 fair
**Presentation:** 2 fair
**Contribution:** 2 fair
**Rating:** 6
**Confidence:** 3

**Summary:**

This paper introduces a unique method to address catastrophic forgetting, drawing inspiration from Causal Effects. Through empirical studies, it highlights the imbalanced causal effect between new and old task data. To address the bias due to the imbalance problem in data between old and new classes, the proposed method strives to equalize the causal impacts of both tasks. Experimental results on CIFAR100 and 5-datasets show that their proposed method outperforms the selected baselines.

**Strengths:**

1. This paper provides a new view on the underlying problem of catastrophic forgetting, drawing attention to the imbalanced causal relationship between data of old and new tasks.
2. The article presents new method named BaCE, utilizing rebalancing methods to alleviate the bias resulting from the imbalanced causal impact between data from both old and new tasks
3. Experimental results on CIFAR100 and 5-datasets are good.

**Weaknesses:**

1. Utilizing a supervised pre-trained model for continual learning may not be an optimal setup, especially when there's an overlap between the pre-training dataset and the continual learning dataset. While it's not strictly prohibited to use a pre-trained model for continual learning, it's preferable to use self-supervised pre-trained models. These models ensure that no label information from the downstream continual learning tasks is leaked.
2. The selected datasets contain limited number of classes. For class-incremental learning, it's recommended to utilize datasets with a broader range of classes, such as ImageNet.

**Questions:**

See above.

---

> ### Author Response · Authors · 2023-11-20
>
> Thanks for the insightful comments. We hope the following responses address your concerns.
>
> **W1**: It's preferable to use self-supervised pre-trained models.
>
> **Response to W1**: Thanks for the suggestion. We have evaluated BaCE on Continual Text Classification and Continual Named Entity Recognition with self-supervised pre-trained BERT. Additionally, we further evaluate on Tiny-ImageNet, CIFAR100, and CIFAR10 with DeiT-S/16 pretrained on 611 classes of ImageNet after 389 classes which are similar or identical to the classes of CIFAR and Tiny-ImageNet are removed. The training settings and experimental results are provided in **"Evaluation on DeiT-S/16 with Less Information Overlapping."** of Appendix D.1.
>
> **W2**: The selected datasets contain a limited number of classes.
>
> **Response to W2**: The revised manuscript adds more experiments on large-scale datasets. We further evaluate BaCE on these large-scale datasets: OmniBenchmark, ImageNet-R, ObjectNet, and Tiny-ImageNet, which contain 300, 200, 200, and 200 classes. The dataset statistics are summarized in Table 5. The experimental results are provided in **"Evaluation on Challenging Datasets."** and **"Evaluation on DeiT-S/16 with Less Information Overlapping."** of Appendix D.1.
>
> Please also refer to **"Improve 1: More experiments on the datasets with less information overlapping and the datasets with a larger number of classes"** in the global response due to the space limitation.

---

### Official Review · Reviewer_ukvZ · 2023-10-31

**Soundness:** 2 fair
**Presentation:** 2 fair
**Contribution:** 2 fair
**Rating:** 3
**Confidence:** 4

**Summary:**

This paper proposed a new method to deal with old and new data imbalance in continual learning based on causal graph. But the experiment settings have several issues.

**Strengths:**

The paper proposed a new method to deal with data imbalance in class-incremental learning. The method is based on a causal graph, which is new.

The proposed method uses a pre-trained model, which is good. In this day and age, many papers still do not use a pre-trained model, which is not understandable.

**Weaknesses:**

1. The data imbalance problem in continual learning is not new. It is weird that the related work on dealing with data imbalance is put in the appendix. Your method of using KD to prevent forgetting is not new.

2. My complaints are about the experiments. First of all, the pre-training data ImageNet 21k already contains most of the classes in cifar100 and 5-datasets, which means that those classes have been learned during pre-training, which is information leak. In continual learning, they aren’t new anymore. That is why less forgetting occurs. The proper way of conducting the experiments is to pre-train using only the data in ImageNet21k with those intersecting classes removed, like this paper: Kim et al. Learnability and Algorithm for Continual Learning. Proceedings of ICML, 2023.

3. It is unfair to compare your model employing a strong pre-trained network with many baselines that don’t use pre-trained network.

4. When comparing your Table 1 and Table 1 in the L2P paper, I found that when L2P performs poorer in a setting than your system, you copied the result from L2P using it as a baseline, but when L2P performs better than your system, you didn't.

5. What does average accuracy mean, average incremental accuracy or average final/last accuracy computed after the last task is learned?

6. Too few datasets are used in the experiments.

7. Many parts of the paper are hard to read.

**Questions:**

See points 1, 4 and 5 in the weakness section.

---

> ### Author Response · Authors · 2023-11-20
>
> Thanks for the insightful comments. We hope the following responses address your concerns.
>
> **W1**: The data imbalance problem in continual learning is not new, and using KD to prevent forgetting is not new.
>
> **Response to W1**: We want to clarify that the confrontational phenomenon (i.e., causal imbalance problem) is different from the class (data) imbalance problem. Both problems indeed come from the imbalance between new and old data. However, the old data is always unavailable or inadequate in the IL scenario. Otherwise, IL is converted into Multi-Task Learning (MTL). The key difference between the class imbalance problem and the causal imbalance problem is that the former only focuses on the **scale of logits** while the latter focuses on the **learning process** of both new and old features and embeddings. We provide more experimental results in the revised manuscript. Please refer to **"Improve 2: More analysis about the causal imbalance problem and the class imbalance problem"** in the global response due to the space limitation.
>
> Furthermore, knowledge distillation (KD) is indeed a common technique in various machine learning problems. In the area of IL, various methods such as BiC, LUCIR, DER++, and CLSER are inspired by the thought of KD. However, how and what to distillate is also an important research question. In this research, we focus on the underexplored causal imbalance problem.
>
> **W2 and W6**: The evaluation datasets have information overlapping with PTMs. Moreover, too few datasets are used in the experiments.
>
> **Response to W2 and W6**:
> Indeed, CIFAR100 has information overlapping with PTMs, and we have discussed it in **"Discussion of the Overlap between Pretraining and Downstream Datasets."** of Appendix D.1 in the original manuscript. The 5-datasets have less information overlapping with PTMs. As mentioned in **"Datasets."** of Appendix D.1, 5-datasets contain samples such as 10 different letters (A-J) and fashion images, which are rare in ImageNet. Some examples can be found in Figure 19 (b).
>
> Thanks for providing the reference [R2-1]. We follow the same experimental setting as in [R2-1] and further evaluate BaCE on Tiny-ImageNet, CIFAR100, and CIFAR10 in the revised manuscript. Furthermore, we also evaluate BaCE on four more challenging datasets: OmniBenchmark, ImageNet-R, ObjectNet, and VTAB. These datasets contain challenging samples that ImageNet pre-trained models cannot handle. Please refer to **"Improve 1: More experiments on the datasets with less information overlapping and the datasets with a larger number of classes"** in the global response due to the space limitation.
>
> **W3**: It is unfair for BaCE to use a strong PTMs while other baselines do not.
>
> **Response to W3**: For all experiments in our paper, we fairly compare BaCE with all other baselines using exactly the same pre-trained checkpoint. We provide the links to these checkpoints in Table 6 and add more descriptions in the revised manuscript to avoid misunderstanding.
>
> **W4**: We didn't compare L2P when BaCE performs poor than L2P.
>
> **Response to W4**: We are not intent to hide the result of L2P [R2-2]. The reason why we do not compare L2P when the buffer size is zero is given in **"Baselines."** of Appendix D.1 in the original manuscript.
>
> From our perspective, **the prompt-based methods such as [R2-2, R2-3] primarily leverage the pretrained knowledge of PTMs instead of learning new knowledge incrementally** because the whole PTMs are frozen. As shown in Table 1 of the L2P paper, L2P does not benefit much from the data replay compared to other methods (with or without replay data).
> In contrast, BaCE and other distillation-based methods (e.g., DER++) finetune the whole PTMs and suffer from catastrophic forgetting more seriously, especially when the buffer size is limited.
> As shown in Table 8,9, the distillation-based methods greatly benefit from replay data, while L2P does not.
> Furthermore, the distillation-based methods, including BaCE, require no additional model components while L2P requires additionally 10$\times$5$\times$768=38.4K and 20$\times$5$\times$768=76.8K parameters for CIFAR100 and 5-datasets respectively.
> Therefore, we believe that these two types of methods are not comparable.
>
>
> **W5**: What does average accuracy mean?
>
> **Response to W5**: The average accuracy in this paper means that the average last accuracy is computed after the last task is learned. We clarified this in the revised manuscript.
>
> **W7**: Many parts of the paper are hard to read.
>
> **Response to W7**: We adjusted the format of Table 1 and the size of Table 8, 9, 13, 16, 19, Figure 12, 13, 14, 19, 20, and 21 to improve the readability.
>
> [R2-1] Learnability and Algorithm for Continual Learning (ICML2023)
>
> [R2-2] Learning to Prompt for Continual Learning (CVPR2022)
>
> [R2-3] Progressive Prompts: Continual Learning for Language Models (ICLR2023)

---

### Official Review · Reviewer_MS7Y · 2023-11-01

**Soundness:** 2 fair
**Presentation:** 2 fair
**Contribution:** 2 fair
**Rating:** 5
**Confidence:** 4

**Summary:**

This paper shows the causal effect of using knowledge distillation and replay memory in class-incremental learning (CIL) setting. The authors pointed out that the main cause of performance degradation is the confrontation phenomenon between old and new samples, and try to resolve this problem with causal graphs. By analyzing the causal path from old data to logits on new classes when performing the knowledge distillation with old and new data. Furthermore, the authors also analyzed the effect of using replay buffer in the causal graph. In the experiment, the proposed method outperforms other baselines.

**Strengths:**

1. By carefully analyzing the causal path in the causal graph, it is clear to see the main cause of confrontation phenomenon between old and new samples.

2. In all experiments, the proposed algorithm achieves higher accuracy than other baselines.

**Weaknesses:**

1. First, the motivation behind constructing the causal graph is similar to [1]. [1] also point out that the path from old samples to intermediate feature can occur the interference, and they also try to resolve this issue. Though figuring out the confrontation phenomenon in CIL may be novel, constructing the causal graph as [1] is quite similar.

2. It would be better to carry out the experiments on large-scale dataset. Since utilizing the pre-training network can also be practical in large-scale dataset setting, showing the effectiveness of BaCE in large scale dataset experiment can strengthen the results.




[1] Hu et. al., "Distilling causal effect of data in class-incremental learning", CVPR, 2021

**Questions:**

1. Why the prediction bias from class imbalance and the confrontation phenomenon are different? Is it wrong that both problem come from imbalanced dataset?

---

> ### Author Response · Authors · 2023-11-20
>
> Thanks for the insightful comments. We hope the following responses address your concerns.
>
> **W1**: The causal graph is similar to [1].
>
> **Response to W1**: The differences between BaCE and DDE [1] have been shown in Appendix C.4 and A.5 in the original version of the manuscript. The causal graph of BaCE contains two subgraphs: one for Effect_{new} and another for Effect_{old}. The subgraph of Effect_{new}is motivated by the causal graph in [1]. However, the motivation behind the causal graph between this research and [1] is different. The subgraph of Effect_{new} in this research is proposed for balancing causal effects when adapting to new classes, while the causal graph in [1] is proposed for retrieving the missing old data effects. Motivated by the issue of causal imbalance, we provide an in-depth analysis of the causalities behind the new classes' logits in Appendix C.3 and define Effect_{new} in Eq (3)(4), which is different from [1].
>
> Furthermore, different from the collider effect in DDE, Effect_{new} estimates the weight of neighbours in Appendix C.3 and thus fundamentally works better than DDE. The experimental result in **"The effect of KNNs in Effect_{new}."** of Appendix D.1 validates the effectiveness of BaCE.
>
>
> **W2**: It would be better to carry out the experiments on large-scale datasets.
>
> **Response to W2**: Thanks for the valuable suggestions. We add more experiments on large-scale datasets in the revised manuscript. We further evaluate BaCE on these large-scale datasets: OmniBenchmark, ImageNet-R, ObjectNet, and Tiny-ImageNet, which contain 300, 200, 200, and 200 classes. The dataset statistics are summarized in Table 5. The experimental results are provided in **"Evaluation on Challenging Datasets."** and **"Evaluation on DeiT-S/16 with Less Information Overlapping."** of Appendix D.1.
>
> **Q1**: Why are the prediction biases from class imbalance and the confrontation phenomenon different? Is it wrong that both problems come from imbalanced datasets?
>
> **Response to Q1**: It is true that both problems come from the imbalance between new and old data. However, in the IL scenario, the old data is always unavailable or inadequate. Otherwise, IL is converted into Multi-Task Learning (MTL). The key difference between the class imbalance problem and the confrontation phenomenon (i.e., causal imbalance problem) is that the former only focuses on the scale of logits while the latter focuses on the learning process of both new and old features and embeddings. We have also clarified the differences in Appendix A.4.
>
> Furthermore, we provide more experimental results in the revised manuscript. Please refer to **"Improve 2: More analysis about the causal imbalance problem and the class imbalance problem"** in the global response due to the space limitation.

---

### Official Review · Reviewer_UgqC · 2023-11-03

**Soundness:** 3 good
**Presentation:** 3 good
**Contribution:** 2 fair
**Rating:** 6
**Confidence:** 4

**Summary:**

This paper proposes to balance causality between old and new tasks with distillation on class incremental learning scenarios. The method uses a variation of the cross entropy loss and the KL loss adapted to whether on which data is available for rehearsal during the sequence learning. Idea is not so novel, but simple and of interest to the community. Experimental results show improvement over compared methods.

**Strengths:**

The proposed distillation method is similar to well-established state-of-the-art methods that tackle the relevant issues from different angles. This point of view and proposed strategy based on causality is of interest and provides good insight into balancing the stability-plasticity trade-off.

The paper is fairly easy to read, the method is well documented and supported by the causality hypotheses defined, and the results support the validity of the method.

**Weaknesses:**

Since the paper focuses on establishing the confrontation phenomenon, but does not provide a method that fully addresses it, the lack of more discussion of the limitations makes it weaker. The point of the paper seems to be juggling between establishing the phenomenon, but not delving as much into it as it could (e.g. more experiments that give insight into what is described in Sec. 4.2), and performing better than the state-of-the-art, but not necessarily tackling what does the proposed method solve actually, in comparison with other similar methods (e.g. where does the performance gain actually come from, is the PTM a huge effect). Some of the interesting insight is relegated to a brief description in the appendix.

IL2M is mentioned as one of the methods that does distillation with a balancing strategy (like BiC and LUCIR), but is not included in the experiments. More analysis on the comparison between these methods that balance only on the logits, and the proposed one would be insightful.

Table 1 is split into two parts and very reduced. Needs to be fixed and more readable. Same goes for most figures, where everything is too squeezed down. Formatting needs to be readable and fixed.

Results in Table 8 of the appendix are a limitation that should be mentioned in the main manuscript. The proposed method can potentially be between 2x to 5x computationally more expensive during training.

**Questions:**

The compared methods for when the buffer is empty is quite limited since EWC and LwF are very classic approaches. Why not compare with more recent rehearsal-free methods? E.g.
- Maintaining Discrimination and Fairness in Class Incremental Learning, CVPR 2020
- Prototype Augmentation and Self-Supervision for Incremental Learning, CVPR 2021
- Class-Incremental Learning via Dual Augmentation, NeurIPS 2021
- Self-Sustaining Representation Expansion for Non-Exemplar Class-Incremental Learning, CVPR 2022
- FeTrIL: Feature Translation for Exemplar-Free Class-Incremental Learning, WACV 2023

When the PTM trained on ImageNet-21k is used for the CIFAR-100 or the 5-datasets, how fair is this comparison? From the ablation in Table 2, it looks as if just freezing parts of the model with the PTM weights is rather competitive with most methods.

Overall, I think it is an interesting submission, of relevance to the field, that could be improved, although it already provides insight into the balancing of the representation biases from a causality perspective.

---

> ### Author Response · Authors · 2023-11-20
>
> Thanks for the insightful comments. We hope the following responses address your concerns.
>
> **W1**: The experiment is not in-depth enough
>
> **Response to W1**: We add more in-depth experiments from the following aspects:
>
> - Exploring the effect of KNNs in BaCE in **“The effect of KNNs in Effect_{new}.”** of Appendix D.1. The results further explain that the KNNs act as a constraint from the teacher model and thus build causal paths from both new and old data when the student models adapt to new classes.
> - Evaluating BaCE on DeiT-S/16 with less information overlapping. The experimental results in **“Evaluation on DeiT-S/16 with Less Information Overlapping.”** of Appendix D.1 shows the superior performance of BaCE.
> - Exploring the feature embedding distance and the probing performance with BiC, IL2M, LUCIR, DER++, and CLSER on four challenging datasets in **“The difference between causal imbalance problem and class imbalance problem.”** of Appendix D.1. Additionally, the confusion matrices of BiC, IL2M, LUCIR and BaCE are provided.
> - Evaluating BaCE on four more datasets: OmniBenchmark, ImageNet-R, ObjectNet, and VTAB. They contain challenging samples and multiple complex realms that ImageNet pre-trained models cannot handle. The experimental results in **“Evaluation on Challenging Datasets.”** of Appendix D.1 shows that BaCE outperforms competitive baselines such as DER++ and CLSER.
>
> **W2**: More analysis on the comparison between BaCE and the methods that balance only on the logits is needed.
>
> **Response to W2**: Please refer to **"Improve 2: More analysis about the causal imbalance problem and the class imbalance problem"** in the global response due to the space limitation.
>
> **W3**: Formatting needs to be readable and fixed.
>
> **Response to W3**: We adjust the format of Table 1 and the size of Table 8, 9, 13, 16, 19, Figure 12, 13, 14, 19, 20, and 21 to improve the readability.
>
> **W4**: A limitation should be included in the main manuscript
>
> **Response to W4**: We added the limitation in the conclusion of the main manuscript.
>
> **Q1**: Why not compare with more recent rehearsal-free methods when the buffer is empty?
>
> **Response to Q1**: Thanks for providing more comparable baselines when the buffer is empty. We will explore their performance in the future. In the current form of the manuscript, we did not compare with them for two reasons:
>
> - The exemplar-based methods achieve superior performance in even challenging scenarios. We compare BaCE with exemplar-based methods such as BiC, LUCIR, IL2M, PODNET, DDE, DER++, Gdumb, CLSER, FOSTER, MEMO, BEEF to show the effectiveness of BaCE.
>
> - The causal graphs show that the confrontational phenomenon still exists (i.e., imbalance causal effects) in the REPLAY setting. We validate the effectiveness of addressing imbalance causal effects by improving upon REPLAY.
>
> Besides, one mentioned method (PASS)[R1-1] is compared in Table 10 of the revised manuscript.
>
> **Q2**: How fair is this comparison when using PTM for the CIFAR-100 or the 5-datasets?
>
> **Response to Q2**: We use the same PTMs for all compared baselines and BaCE. Furthermore, we add more experiments on scenarios with less information overlapping. Please also refer to **"Improve 1: More experiments on the datasets with less information overlapping and the datasets with a larger number of classes"** in the global response due to the space limitation.
>
> The IL performance on downstream datasets is indeed positively correlated with the power of PTMs. Recently, many prompt-based methods, such as L2P [R1-2] and Progressive Prompt [R1-3], have been proposed. However, these partially fine-tuning methods (including the frozen baselines in Table 3) primarily leverage the pretrained knowledge of PTMs instead of learning new knowledge incrementally. As shown in Table 8, L2P does not benefit much from the data replay compared to other methods. In other words, partially fine-tuning methods have superior performance, especially when the replay data is limited. In contrast, BaCE and other distillation-based methods can be applied to more IL scenarios with different backbones, datasets, and buffer sizes.
>
> [R1-1] Prototype Augmentation and Self-Supervision for Incremental Learning (CVPR2021)
>
> [R1-2] Learning to Prompt for Continual Learning (CVPR2022)
>
> [R1-3] Progressive Prompts: Continual Learning for Language Models (ICLR2023)

---

> > ### Comment · Reviewer_UgqC · 2023-11-22
> >
> > Thanks to the authors for their detailed responses. I appreciate the changes and the added experiments, and would recommend providing a better representation of the results for the pre-trained models without dataset overlap on the main manuscript. I think this is one of the most relevant key points and needs to be properly addressed in the main manuscript and not in the appendix. The results of Table 14 on the effects of the KNN address the issue I raised, although seem to be very close among all the variants, while the proposed BACE being alone at the top seems odd. I would have expected a larger variety among the other ablated versions at least.
> > All in all, I think that the authors have addressed most of my concerns and I would lean towards acceptance. However, I do still believe that the article would be more interesting if the focus was more on the solving and analyzing the confrontation phenomenon details.

---

> > > ### Author Response · Authors · 2023-11-23
> > >
> > > Thank you for the comments.
> > > -   We will move the results without dataset overlap to the main manuscript in the next version.
> > > -   The results of other baselines on the four challenging datasets used in Table 14 are provided in Table 12. The results among variants/baselines are close because the four datasets are more challenging than CIFAR100. As shown in Table 12, BaCE outperforms other baselines, and the accuracy of BaCE approaches the probing accuracy on OmniBenchmark, ImageNet-R, and ObjectNet.
> > > -   We will continue to explore the confrontation phenomenon to gain more insights.

---

### Author Response · Authors · 2023-11-20
**Response to All Reviewers**

We thank all reviewers for their valuable comments. We improved the manuscript according to the questions and suggestions. The modified parts are highlighted in teal. We also uploaded the source code and the additional results in the supplementary material.

**Improve 1: More experiments on the datasets with less information overlapping and the datasets with a larger number of classes**

To mitigate the information overlapping between pretraining data and the datasets for incremental learning, we follow the experimental setting in [R1] to further evaluate the proposed method. Specifically, we use the same DeiT-S/16 [R2] checkpoint, which is trained using 611 classes of ImageNet after removing 389 classes which are similar or identical to the classes of CIFAR and Tiny-ImageNet. The experimental results on Tiny-ImageNet, CIFAR100, and CIFAR10 outperform existing methods. The training settings and experimental results are provided in **“Evaluation on DeiT-S/16 with Less Information Overlapping.”** of Appendix D.1.

Furthermore, we follow the experimental setting in [R3], use ViT-B/16-IN21K as the backbone, and evaluate models on four challenging datasets: OmniBenchmark, ImageNet-R, ObjectNet, and VTAB. These datasets contain challenging samples and multiple complex realms that ImageNet pre-trained models cannot handle. The experimental results on the four challenging datasets show that BaCE outperforms a series of competitive methods. The training settings and experimental results are provided in **“Evaluation on Challenging Datasets.”** of Appendix D.1.


**Improve 2: More analysis of the causal imbalance problem and the class imbalance problem**

The causal imbalance problem focuses on the learning process of both new and old features and embeddings, while the class imbalanced problem only focuses on the scale of logits.
We compare BaCE and three classical methods for class imbalance problems, including BiC, IL2M, and LUCIR, on the four challenging datasets.
We find that addressing the causal imbalance problem significantly reduces the feature embedding distance between new and old classes.
More importantly, it improves the probing accuracy over existing methods designed for the class imbalance problem.
**It indicates that only balancing the scale of logits (e.g., BiC and IL2M) may not necessarily enhance the feature learning process.**
In contrast, BaCE is proposed for addressing causal imbalance problem and pursues causal-balanced learning for each class, thereby achieving superior probing performance in IL.

Besides, we visualize the confusion matrices of BaCE, BiC, LUCIR, and IL2M.
The confusion matrices show that BiC shows the most balanced predictions, followed by BaCE and IL2M, and finally, LUCIR.
However, BiC and IL2M make more errors in predicting old classes and perform lower than BaCE.
We speculate that the reason behind this is that the old rehearsal data does not reflect the true distribution of the old data, and pursuing a balanced prediction between the rehearsal data and new data may hurt the representation ability during IL.

Furthermore, we learn two additional parameters, as in BiC, to balance the prediction of BaCE, and we find that the performance increases slightly.
It indicates that the causal imbalance problem and class imbalance problem are not conflicting.
The technique for the class imbalance problem can be combined with BaCE to improve the performance further.
The training settings and experimental results are provided in **“The difference between causal imbalance problem and class imbalance problem.”** of Appendix D.1.
The confusion matrices are provided in the supplementary material.

**In summary, analyzing the causalities in CIL helps us to clarify the reason behind forgetting. The causal perspective helps us design algorithms that better maintain the model representation ability in IL. In contrast, the methods for the class imbalance problem superficially balance the scale of logits, and thus, they are difficult to guide the model to learn better features.**

[R1] Learnability and Algorithm for Continual Learning (ICML2023)

[R2] Training data-efficient image transformers distillation through attention (ICML2021)

[R3] Revisiting Class-Incremental Learning with Pre-Trained Models: Generalizability and Adaptivity are All You Need (arxiv)

---

### Meta-Review · Area_Chair_Hvys · 2023-12-08

**Metareview:**

(a) Summarize the scientific claims and findings of the paper based on your own reading and characterizations from the reviewers.
- The authors discover a new interpretation of forgetting in class-incremental continual learning.
- It is related to the difference in class representation in the encoder and in the classification head(s). In short, during continual learning, the the representation of old classes of the encoder and classifier becomes unaligned (hence forgetting).
- The authors develop an mdoel of this phenomenon through the terminology of causality
- With this initial finding, they develop a method (BaCE) and study its behavior empircally

(b) What are the strengths of the paper?
- A novel causal analysis of the effects of data imbalance in class-incremental settings
- The analysis leads to the new method

(c) What are the weaknesses of the paper? What might be missing in the submission?
- Relation to class imbalance. Other works have studied various class imbalance problems. The exact distinction between class imbalance and the "confrontational phenomenon" of this paper requires clarifications (even after reading the subsection in the appendix about it). In other words, the causality perspective seems to lead to the desiderata of better balancing new and old tasks. This seems like a setting that has been studied before under the umbrella of class imbalance.
- Information leakage. When doing continual learning from (supervised) pre-trainted models, information from the pre-train data can leak and bias the continual learning results. The authors recognize this possibility and, in their response, provide new results using a "standard" protocol to remove (or at least reduce) leakage. The resulting performance gains of their approach then seem marginal and perhaps even not significant (see Table 10, 11 an Appendix D.1).
- Fairness of comparisons with L2P. Even in the revised version, the proposed method is only compared to L2P in the large-replay regime. The authors justify this choice, but at least providing the results with smaller replay buffers would be helpful.

**Justification For Why Not Higher Score:**

The reviewers did not reach a consensus, and the paper remains borderline according to their evaluation.

Reviewer ukvZ did provide feedback privately to myself and the other reviewers after your response. Their main arguments were related to data leakage, unclear differentiation with class imbalance work, and unfair comparison with L2P (and missing with its "successor" DualPrompt from ECCV-2022).

The authors did try to clarify the difference with the work on imbalance classes. Further, the justification that leads to this finding in this work difers from others and as such might be valuable to the research community. Overall, the reviewers all agreed about the value of this finding. That is the main strength of the paper.

The arguments with respect to the empirical evaluation (reviewer MS7Y also has seemingly unresolved concerns) are more important. I don't find it critical to obtain state-of-the-art results, but the critic from the reviewers imply a lack of rigour in the reported results.

In my view, this remains a borderline paper, but one that would clearly benefit from another round of reviewing to provide a thorough analysis of the new results. I am sorry that I cannot recommend acceptance at this stage.

**Justification For Why Not Lower Score:**

N/A

---

### Decision · Program_Chairs · 2024-01-16

Reject